

# Reynolds-number dependence of turbulence enhancement on collision growth

Ryo Onishi[1] and Axel Seifert[2]

[1]Center for Earth Information Science and Technology, Japan Agency for Marine-Earth Science and Technology, 3173-25 Showa-machi, Kanazawa-ku, Yokohama Kanagawa 236-0001 Japan
[2]Deutscher Wetterdienst, Offenbach, Germany

*Correspondence to:* Ryo Onishi (onishi.ryo@jamstec.go.jp)

**Abstract.** This study investigates the Reynolds-number dependence of turbulence enhancement on the collision growth of cloud droplets. The Onishi turbulent coagulation kernel proposed in Onishi et al. (2015) is updated by using the direct numerical simulation (DNS) results for the Taylor-microscale-based Reynolds number ($Re_\lambda$) up to 1,140. The DNS results for particles with a small Stokes number ($St$) show a consistent Reynolds-number dependence of the so-called clustering effect with the locality theory proposed by Onishi et al. (2015). It is confirmed that the present Onishi kernel is more robust for a wider $St$ range and has better agreement with the Reynolds-number dependence shown by the DNS results. The present Onishi kernel is then compared with the Ayala-Wang kernel (Ayala et al. (2008a); Wang et al. (2008)). At low and moderate Reynolds numbers both kernels show similar values except for $r_2 \sim r_1$, for which the Ayala-Wang kernel shows much larger values due to its large turbulence enhancement on collision efficiency. A large difference is observed for the Reynolds-number dependences between the two kernels. The Ayala-Wang kernel increases for the autoconversion region ($r_1, r_2 < 40\ \mu$m) and for the accretion region ($r_1 < 40\ \mu$m and $r_2 > 40\mu$m; $r_1 > 40\ \mu$m and $r_2 < 40\mu$m) as $Re_\lambda$ increases. In contrast, the Onishi kernel decreases for the autoconversion region and increases for the rain-rain self-collection region ($r_1, r_2 > 40\ \mu$m). Stochastic collision-coalescence equation (SCE) simulations are also conducted to investigate the turbulence enhancement on particle size evolutions. The SCE with the Ayala-Wang kernel (SCE-Ayala) and that with the present Onishi kernel (SCE-Onishi) are compared with results from the Lagrangian Cloud Simulator (LCS, Onishi et al. (2015)), which tracks individual particle motions and size evolutions in homogeneous isotropic turbulence. The SCE-Ayala and SCE-Onishi kernels show consistent results with the LCS results for small $Re_\lambda$. The two SCE simulations, however, show different Reynolds-number dependences, indicating possible large differences in atmospheric turbulent clouds with large $Re_\lambda$.

## 1 Introduction

Several mechanisms have been proposed to explain the rapid growth of cloud droplets, which often result in fast rain initiation in the early stages of cloud development. Examples of these mechanisms include the turbulence-enhanced collision rate of cloud droplets (Falkovich and Pumir (2007); Grabowski and Wang (2013)), turbulent entrainment (Blyth (1993); Krueger et al. (1997)), giant cloud condensation nuclei (Yin et al. (2000); Van Den Heever and Cotton (2007)), and turbulent dispersions of cloud droplets (Sidin et al. (2009)). The first mechanism, which has received the most attention, has led to extensive research





on particle collisions in turbulence (e.g., Sundaram and Collins (1997); Wang et al. (2000); Saw et al. (2008); Onishi et al. (2009); Dallas and Vassilicos (2011)).

One direction taken by the research in this area is the simulation of collisional growth by solving the stochastic collision-coalescence equation (SCE). Such research relies on accurate collision-coalescence models, which consist of models for the
collision kernel $K_c(r_1, r_2)$ (where $r_i$ is the particle radius), the collision efficiency $E_c(r_1, r_2)$, and the coalescence efficiency $E_{coal}(r_1, r_2)$. To consider the influence of turbulence, several turbulent collision models have been proposed. Saffman and Turner (1956) analytically derived a collision kernel model for particles with no inertia or with a very small Stokes number ($St = \tau_p/\tau_\eta$, where $\tau_p$ is the particle relaxation time and $\tau_\eta$ is the Kolmogorov time), while Abrahamson (1975) derived a model for $St \gg 1$. For moderate Stokes numbers, i.e., $St \sim 1$, one difficulty is the preferential motion of inertial particles.
Inertial particles preferentially cluster in regions of low vorticity and high strain if $St \ll 1$ (Maxey (1987)), and cluster in a way that mimics the clustering of zero-acceleration points by the sweep-stick mechanism if $1 \lesssim St \lesssim \tau_p/T_I$, where $T_I$ is the integral timescale of the turbulence (Coleman and Vassilicos (2009)). This matters because clustering increases the mean collision rate (Sundaram and Collins (1997)). To quantify the clustering due to the preferential concentration effect a model is formulated for finite-inertial particles. However, the model requires several empirical parameters that should be determined
from reference data, e.g., results from a direct numerical simulation (DNS).

One serious problem is that the Reynolds-number dependence of turbulent collisions has not yet been clarified. Actually, many authors ignore the Reynolds-number dependence and assume a constant collision kernel regardless of the Reynolds number (e.g., Saffman and Turner (1956); Derevyanko et al. (2008); Zaichik and Alipchenkov (2009)) or assume a convergence to a constant collision kernel with increasing Reynolds number (e.g., Ayala et al. (2008a)). Onishi et al. (2013) observed that
the clustering effect, and consequently the collision kernel, decreases as the Taylor-microscale-based Reynolds number ($Re_\lambda$) increases for $St$=0.4. Onishi and Vassilicos (2014) later clarified that the Reynolds-number dependence of the clustering effect for $1/3 \lesssim St \lesssim 1$ is due to internal intermittency of the turbulence. Because a robust theoretical model for turbulent collision kernels is not yet available, we need empirical models for the investigation of turbulence enhancement on cloud development. As an example, the Ayala-Wang kernel (Ayala et al. (2008a); Wang et al. (2008)) is a widely used turbulent kernel model.

Recently, Onishi et al. (2015) proposed an empirical kernel model based on DNS data for the wide range of $49 \leq Re_\lambda \leq 530$, where $Re_\lambda$ is the Taylor-microscale-based Reynolds number. Onishi et al. (2015) also conducted stochastic and direct collision simulations to investigate the turbulence enhancement on drop size evolution. They investigated the energy dissipation ($\epsilon$) dependence for the range of $100 \leq \epsilon \leq 1{,}000$ cm$^2$/s$^3$ and the $Re_\lambda$ dependence for the range of $66 \leq Re_\lambda \leq 206$. The results showed good agreement of the $\epsilon$ dependence between the stochastic simulations with the Ayala-Wang and Onishi kernels, but
a significant discrepancy for the $Re_\lambda$ dependence between the two kernels. The discrepancy in $Re_\lambda$ dependence may become a critical issue for cloud simulations because $Re_\lambda$ is typically as large as O($10^{3-4}$) in atmospheric turbulent clouds. However, Onishi et al. (2015) did not provide a detailed discussion on the difference of the Ayala-Wang and Onishi kernels in $Re_\lambda$ dependence.

This study, therefore, aims to compare the Ayala-Wang and Onishi kernels by focusing on their $Re_\lambda$ dependence. First,
the Onishi kernel is updated by using the reference collision statistics obtained by the DNS for $Re_\lambda$ up to 1,140. The Ayala-





Wang and the present Onishi kernel values are compared in detail. The SCE simulations with the Ayala-Wang and Onishi kernels are also compared with each other and with the reference results from the Lagrangian Cloud Simulator (LCS, Onishi et al. (2015)), which tracks individual particle motions and size evolutions in homogeneous isotropic turbulence. The collision growth simulation with the LCS is conducted for $Re_\lambda$ up to 333.

## 2   Turbulent Coagulation Kernel Models

### 2.1   Turbulent coagulation kernel

The geometric collision frequence per unit volume between particles with radius $r_1$ and those with radius $r_2$, $N_c(r_1, r_2)$, is expressed by the geometric collision kernel $K_c(r_1, r_2)$ as

$$N_c(r_1, r_2) = K_c(r_1, r_2)\, n_{p,1} n_{p,2}, \tag{1}$$

where $n_{p,i}$ is the number density of particles with radius $r_i$. The coagulation kernel $K_{coag}$ can be expressed by the combination of the geometric collision kernel, collision efficiency $E_c$ and coalescence efficiency $E_{coal}$ as

$$K_{coag}(r_1, r_2) = E_{coal}(r_1, r_2)\, E_c(r_1, r_2)\, K_c(r_1, r_2). \tag{2}$$

The gravitational collision kernel describes the collisions due to the settling velocity difference in the form of

$$K_{c,grav}(r_1, r_2) = \pi R_{12}^2 |V_{\infty 1} - V_{\infty 2}|, \tag{3}$$

where $R_{12}\,(= r_1 + r_2)$ is the collision radius and $V_{\infty i}$ is the gravitational particle settling velocity. Turbulence enlarges the geometric collision kernel, i.e., the turbulent geometric kernel $K_{c,turb}$ is larger than $K_{c,grav}$. Turbulence also enhances the coagulation kernel through enlarging $E_c$. The turbulence enhancement on the collision efficiency, $\eta_E$, is defined as

$$\eta_E(r_1, r_2) = \frac{E_c(r_1, r_2)\,[T]}{E_c(r_1, r_2)\,[NoT]}, \tag{4}$$

where [T] and [NoT] indicate the turbulent flow case and the stagnant (non-turbulent) flow case, respectively.

It had been difficult to confidently discuss the collision efficiency in a turbulent flow until Ayala et al. (2007) developed a reliable superposition method, which iteratively solves the Stokes disturbance flows for a many-particle system. That superposition method is, however, computationally expensive due to its iteration procedure. Onishi et al. (2013) later developed a less costly method, named the binary-based superposition method (BiSM), which has been adopted in the LCS (Onishi et al. (2015)). BiSM assumes that interactions via three or more particles are negligible. This dramatically reduces the computational 25  cost but maintains reliability as long as the particle number concentration is small, as observed in atmospheric clouds.





Sundaram and Collins (1997) showed, by means of a DNS, that the preferential concentration of inertial particles, the so-called clustering effect, increases the collision frequency. The clustering effect is expressed in the spherical formulation derived by Wang et al. (1998) as

$$K_c(r_1, r_2) = 2\pi R_{12}^2 \langle |w_r(x = R_{12})| \rangle g_{12}(x = R_{12}),$$ (5)

where $\langle \cdots \rangle$ denotes an ensemble average, $|w_r(x = R_{12})|$ ($|w_r|$ hereafter) is the radial relative velocity at contact separation, and $g_{12}(x = R_{12})$ ($g_{12}$ hereafter) is the radial distribution function at contact separation and represents the clustering effect.

## 2.2 Ayala model

Ayala et al. (2008a) provided a parameterization for the turbulent geometric collision kernel of finite-inertia sedimenting droplets by proposing an empirical model for $g_{12}$ in addition to a theoretical model for $\langle |w_r| \rangle$.

By following the expression by Chun et al. (2005), the clustering effect for a monodisperse suspension of sedimenting droplets is expressed as

$$g_{11} = \left(\frac{\eta}{r}\right)^{C_1},$$ (6)

where $\eta$ is the Kolmogorov length. $C_1$ is a function of $St$, $Re_\lambda$ and the non-dimensional parameter for gravity $V_\infty/v_\eta$ with the Kolmogorov velocity $v_\eta$. This parameterization was extended for a bidisperse system in a manner similar to that in Chun et al. (2005):

$$g_{12} = \left(\frac{\eta^2 + r_d^2}{r_L^2 + r_d^2}\right)^{C_1/2},$$ (7)

where $r_L$=max($r_1,r_2$) and $C_1$ follow the same expression for the monodisperse case at $St_{\max}$=max($St_1,St_2$)=$St(r_L)$, and $r_d$ is a length scale of the acceleration diffusion experienced by the particles. When two particles in a pair are two different sizes, any fluid acceleration or gravity will induce a relative velocity. This effect yields a diffusion-like process in the system and tends to smooth out inhomogeneities in the particle pair concentration. Thus, $r_d$ is larger for larger $|St_1 - St_2|$ for the bidisperse case and a monodisperse suspension form is recovered for the case $r_d \ll r_L$. It should be noted for the discussion in subsection 4.4 that the $g_{12}$ model was designed to show maximum clustering at $St \sim 1$ and a higher droplet clustering for larger $Re_\lambda$ (Ayala et al. (2008b)).

In addition to the empirical $g_{12}$ model, Ayala et al. (2008a) developed a theory for $\langle |w_r| \rangle$ that is applicable to inertial droplets sedimenting under gravity in a turbulent flow. The basic assumption was that the droplet relative trajectory is mostly determined by gravitational sedimentation. Following Dodin and Elperin (2002), they decomposed the radial relative velocity (between two



particles falling under gravity in a homogeneous isotropic turbulent flow) into a random part $\xi$ caused by turbulent fluctuations and a deterministic part $h$ due to gravity:

$$w_r(\phi) = \xi(\phi) + h(\phi), \tag{8}$$

where the angle of contact, $\phi$, is measured from the gravity axis. The random variable $\xi(\phi)$ is assumed to be normally distributed with a standard deviation $\sigma(\phi)$.

Using $\sigma(\phi = 90°)$ to approximate $\sigma(\phi)$, they obtained

$$\langle |w_r| \rangle = \sqrt{\frac{2}{\pi}} \left( \sigma^2 + \frac{\pi}{8} (\tau_{p,1} - \tau_{p,2})^2 g^2 \right)^{1/2}, \tag{9}$$

where $\sigma$ is expressed in terms of $\tau_{p,i}$, $V_{\infty i}$ and flow parameters $u_{rms}$ (the rms of the velocity fluctuations) in terms of $\epsilon$ and $Re_\lambda$.

## 2.3 Onishi model

### 2.3.1 model for $g_{12}$

Onishi et al. (2015) proposed an original model for the clustering effect in monodisperse systems.

$$g_{11} - 1 = \begin{cases} A_1 St^2 & (\equiv y_1) \text{ (for } St < St_a) \\ A_2 Re_\lambda St^{-2} & (\equiv y_2) \text{ (for } St_a \leq St) \end{cases}, \tag{10}$$

where $A_1$ and $A_2$ were empirically determined to be 110 and 0.38, respectively. The regime boundary $St_a$ is $(A_2/A_1)^{1/4} Re_\lambda^{1/4}$. A tanh smoothing function, $z_a$, was employed to connect the two formulations in the equation as

$$g_{11} - 1 = H(St - St_a) y_1 z_a^\alpha + H(St_a - St) y_2 (1 - z_a)^\alpha. \tag{11}$$

(Note that the Heaviside function was missing in Onishi et al. (2015).) Here,

$$z_a(St) = \frac{1}{2} \left( 1 - \tanh \frac{\log_{10} St - \log_{10} St_a}{C_a} \right), \tag{12}$$

where $C_a$ is parameterized as

$$C_a = a_c Re_\lambda^{b_c}. \tag{13}$$



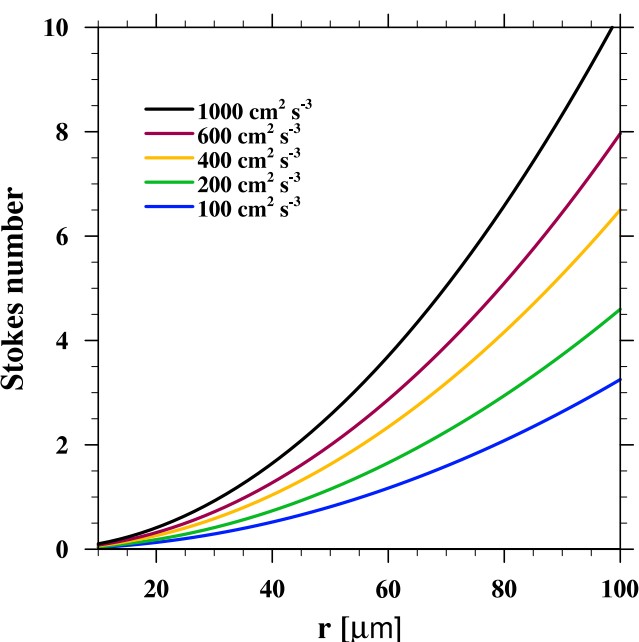

**Figure 1.** Stokes number against the particle radius for various energy dissipation rates.

in Eq. (11), $\alpha$ is parameterized as

$$\alpha = \log_2 a_\alpha Re_\lambda^{b_\alpha}. \tag{14}$$

Onishi et al. (2015) determined the optimal values for the abovementioned empirical coefficients (i.e., $A_1$, $A_2$, $a_c$, $b_c$, $a_\alpha$, and $b_\alpha$) based on the dataset in Onishi et al. (2013); Onishi and Vassilicos (2014) for $St \leq 1$.

5    If we limited the discussion for the autoconversion regime, i.e., $r < 40$ $\mu$m, the range $St \leq 1$ would be enough for the typical energy dissipation rate $\epsilon \leq 1000$ cm$^2$/s$^3$ observed in atmospheric turbulent clouds. However, as clearly shown in Figure 1, $St$ can be as large as 10 for $r = 100$ $\mu$m and $\epsilon = 1000$ cm$^2$/s$^3$. That is, in the discussion on the accretion process that describes the conversion from cloud to rain due to rain drops collecting cloud droplets, we need to deal with $St > 1$ as well.


| | $A_1$ | $A_2$ | $a_c$ | $b_c$ | $a_{c2}$ | $b_{c2}$ | $a_\alpha$ | $b_\alpha$ |
|---|---|---|---|---|---|---|---|---|
| Onishi et al. (2015) | 110 | 0.38 | 0.060 | 0.30 | - | - | 0.26 | 0.50 |
| present | 110 | 0.32 | 0.046 | 0.36 | 0.094 | 0.25 | 0.23 | 0.50 |

**Table 1.** Parameter values for $g_{11}$ model.

Hence, this study modifies the parameterization in the original Onishi kernel to obtain better overall matching for a wider range of $St$. After trial and error, we finally obtained a modification of the form of $C_a$ as

$$C_a = \min\left(a_c Re_\lambda^{b_c}, a_{c2} Re_\lambda^{b_{c2}}\right).$$ (15)

We confirmed that this form with $a_c = 0.046$, $b_c = 0.36$, $a_{c2} = 0.094$, and $b_{c2} = 0.25$ leads to an improvement, as shown later in subsection 4.2. The updated coefficients are summarized in Table 1.

To determine the clustering effect for bidisperse systems, the empirical formulation proposed by Zhou et al. (2001) is employed:

$$g_{12} = 1 + \rho_{12}\left(g_{11} - 1\right)^{1/2}\left(g_{22} - 1\right)^{1/2},$$ (16)

where $\rho_{12}$ is an empirical function of $St_{\max}$.

### 2.3.2 Model for $\langle|w_r|\rangle$

Onishi et al. (2015) employed the model of Wang et al. (2000) for $\langle|w_r|\rangle$, which was based on the model by Kruis and Kusters (1997), as

$$\langle|w_r|\rangle = \left[\frac{2}{\pi}\left(w_{shear}^2 + w_{accel}^2\right)\right]^{1/2},$$ (17)

$$w_{shear}^2 = \frac{R^2\epsilon}{15\nu},$$ (18)

$$w_{accel}^2 = \frac{1}{3}C_w\left(St_{\max}\right)f_{KK},$$ (19)

where $\nu$ is the kinematic viscosity and $C_w(St_{\max}) = 1 + 0.6\exp\left[-(St_{\max} - 1)^{1.5}\right]$. The formulation of $f_{KK}$ was proposed by Kruis and Kusters (1997) as





$$
\begin{aligned}
f_{KK} &= \frac{\gamma u_{rms}^2}{\gamma - 1} \left\{ (\theta_1 + \theta_2) - \frac{4\theta_1 \theta_2}{(\theta_1 + \theta_2)} \left[ \frac{1 + \theta_1 + \theta_2}{(1 + \theta_1)(1 + \theta_2)} \right]^{1/2} \right\} \\
&\times \left[ \frac{1}{(1 + \theta_1)(1 + \theta_2)} - \frac{1}{(1 + \gamma\theta_1)(1 + \gamma\theta_2)} \right],
\end{aligned}
\tag{20}
$$

where $\theta_i = \tau_{p,i}/T_L$ with $T_L$ as the Lagrangian integral time, and $\gamma = 0.183 u_{rms}^2/(\epsilon\nu)^{1/2}$. In the equation, $\theta_i$ shows the relative particle relaxation time to the particle-flow interaction time. Note that this $\langle |w_r| \rangle$ parameterization is for non-sedimenting

droplets.

Onishi et al. (2009) concluded that gravitational sedimentation does not significantly influence turbulent collisions of cloud droplets. However, for this study, which extends the discussion to the small rain drop regime, the gravitational sedimentation cannot be ignored. Therefore, this study introduces a simple modification to make the model applicable to sedimenting droplets by considering the mechanism in which the gravitational settling shortens the interaction time of droplets with eddies Onishi

et al. (2009). The enlargement of the relative particle relaxation time by gravity is modeled as

$$
\theta_{i,sed} = \sqrt{\frac{3(1 - f(\theta_i)) + s_v^2}{3(1 - f(\theta_i))}} \theta_i,
\tag{21}
$$

where $f(\theta)$ is defined as the ratio of the particle velocity fluctuation to the flow velocity fluctuation, i.e., $f(\theta) = v_p'^2/u_{rms}^2$, and $s_v = V_{p,\infty}/u_{rms}$ is a non-dimensional parameter quantifying the influence of sedimentation. By replacing $\theta_i$ in Eq. (20) by $\theta_{i,sed}$, we obtain the radial relative velocity for droplets with gravitational sedimentation, $\langle |w_r| \rangle_{turb,sed}$.

The above simple treatment is not yet complete. Ayala et al. (2008a) suggested the following two contributions of gravitational sedimentation on $\langle |w_r| \rangle$; (i) gravity reduces the interaction time of droplets with turbulent eddies, and therefore the variance of particle velocities is reduced, and (ii) gravity also decreases the correlation coefficient. The second contribution is missing in the present simple treatment. Nonetheless, since the present treatment leads to an improvement in the turbulent coagulation kernel, as shown in subsection 4.3, this study adopts this simple treatment and leaves more robust treatment to

future work.

The turbulent collision kernel formulated from the above $g_{12}$ and $\langle |w_r| \rangle_{turb,sed}$ does not include the collision contribution due to the settling velocity difference. To include the contribution of the settling velocity difference, the following simple formulation was employed to obtain the total collision kernel.

$$
K_{c,total}(r_1, r_2) = \left( K_{c,turb}^2(r_1, r_2) + K_{c,grav}^2(r_1, r_2) \right)^{1/2}
\tag{22}
$$

Here, $K_{c,turb}$ denotes the turbulent collision kernel obtained by $K_{c,turb} = 2\pi R_{12}^2 \langle |w_r| \rangle_{turb,sed} g_{12}$. This simple form is exact if no clustering ($g_{12} = 1$) occurs and $\langle |w_r| \rangle_{turb,sed}$ and $\langle |w_r| \rangle_{grav}$ follow Gaussian distributions.



### 2.3.3 Turbulent enhancement on collision efficiency

Onishi et al. (2015) employed the collision efficiency values of Pinsky et al. (2001) ($E_{c,PKS01}$ hereafter) and $\eta_E$ tabulated in Wang et al. (2008). These tabulated values spanned a relatively small range of particle sizes: the sizes of collector droplets ($r_1$) were 20, 30, and 50 $\mu$m and the size ratios ($r_2/r_1$) were from 0.167 to 0.90. Later, Wang and Grabowski (2009) tabulated the preliminary values of the enhancement factor for a wider range of droplet sizes: $r_1$=20, 30, 40, and 50 $\mu$m and $r_2/r_1$ from 0.0 to 1.0. Note that the data for $r_2/r_1$=0.0 were simply set to the values for $r_2/r_1$=0.0835. It should also be noted that Wang and Grabowski (2009) tabulated the enhancement factors against the Hall collision efficiency ($E_{c,Hall}$ hereafter, Hall (1980)). Unfortunately, inconsistencies exist between the two collision efficiency models. We found differences that are sometimes much larger than 10% of the mean between $E_{c,PKS01}$ and $E_{c,Hall}$, particularly for small and large $r_2/r_1$ ratios, i.e., for $r_2/r_1 \sim 0$ and $\sim 1$. These differences should be carefully compensated in $\eta_E$. Wang and Grabowski (2009) tabulated the enhancement on $E_{c,Hall}$, $\eta_E^{\#Hall}$. In fact, we observed an overestimate in turbulent enhancement on the autoconversion rate when we used $\eta_E^{\#Hall}$ for the SCE simulation with $E_{c,PKS01}$. For Table 2, we calculated $\eta_E$ against $E_{c,PKS01}$ ($\eta_E^{\#PKS01}$) from $\eta_E^{\#Hall}$ as

$$\eta_E^{\#PKS01}(r_1, r_2) = \frac{E_{c,Hall}(r_1, r_2)}{E_{c,PKS01}(r_1, r_2)} \eta_E^{\#Hall}(r_1, r_2). \tag{23}$$

Following Wang and Grabowski (2009), this study simply sets the values for $r_1 \leq 20\ \mu$m to those at $r_1$=20 $\mu$m, and similarly the values at $r_1$=60 $\mu$m to those at $r_1$=50 $\mu$m. The factor is set to unity for $r_1$=100 $\mu$m and larger. Also, following Seifert et al. (2010), for $100 \leq \epsilon \leq 600$ cm$^2$/s$^3$, this study linearly interpolates/extrapolates between the values of $\eta_E^{\#PKS01}$ at $\epsilon$=100 cm$^2$/s$^3$ and at $\epsilon$=400 cm$^2$/s$^3$. For $\epsilon$>600 cm$^2$/s$^3$ the extrapolated values at $\epsilon$=600 cm$^2$/s$^3$ are used for $\eta_E^{\#PKS01}$.

## 3 Direct Numerical Simulations

### 3.1 Computational methods

We now solve the three-dimensional continuity and Navier-Stokes equations for incompressible flows:

$$\nabla \cdot \mathbf{U} = 0, \tag{24}$$

$$\frac{\partial \mathbf{U}}{\partial t} + (\mathbf{U} \cdot \nabla)\mathbf{U} = -\frac{1}{\rho}\nabla p + \nu \nabla^2 \mathbf{U} + \mathbf{F}(\mathbf{x}, t). \tag{25}$$

The kinematic viscosity $\nu$ is set to $1.5 \times 10^{-5}$ m$^2$s$^{-3}$, which is the value for atmospheric air at 1 atm and 298 K. The last term on the right-hand side represents the external forcing needed to achieve a statistically steady state. This study employs reduced-communication forcing (RCF) (Onishi et al. (2011)), which is suitable for massively parallel finite-difference models



| $r_2/r_1$ | $r_1 =$ 20$\mu$m | 30$\mu$m | 40$\mu$m | 50$\mu$m | $r_2/r_1$ | $r_1 =$ 20$\mu$m | 30$\mu$m | 40$\mu$m | 50$\mu$m |
|---|---|---|---|---|---|---|---|---|---|
| 0.0 | 1.74 | 1.77 | 1.49 | 1.21 | 0.0 | 4.98 | 3.59 | 2.52 | 1.45 |
| 0.1 | 5.26 | 3.55 | 2.31 | 1.65 | 0.1 | 10.7 | 5.45 | 3.13 | 1.86 |
| 0.2 | 2.67 | 0.742 | 1.29 | 1.04 | 0.2 | 4.03 | 0.879 | 1.51 | 1.20 |
| 0.3 | 1.75 | 0.733 | 1.15 | 1.04 | 0.3 | 2.08 | 0.758 | 1.22 | 1.15 |
| 0.4 | 0.995 | 0.953 | 1.11 | 1.06 | 0.4 | 1.05 | 0.973 | 1.14 | 1.100 |
| 0.5 | 0.955 | 1.06 | 1.03 | 1.03 | 0.5 | 0.751 | 1.19 | 1.10 | 1.05 |
| 0.6 | 0.730 | 1.11 | 1.00 | 1.03 | 0.6 | 0.832 | 1.29 | 1.10 | 1.07 |
| 0.7 | 0.701 | 1.07 | 0.983 | 0.991 | 0.7 | 0.929 | 1.29 | 1.10 | 1.02 |
| 0.8 | 1.01 | 1.18 | 1.06 | 1.01 | 0.8 | 1.42 | 1.41 | 1.21 | 1.09 |
| 0.9 | 1.63 | 1.81 | 1.34 | 1.31 | 0.9 | 3.94 | 2.19 | 1.51 | 1.34 |
| 1.0 | 29.2 | 6.10 | 2.89 | 3.14 | 1.0 | 22.6 | 5.47 | 2.18 | 1.88 |
| | (a) | | | | | (b) | | | |

Table 2. Enhancement factor for the Pinsky collision efficiency (PKS01), $\eta_E^{\#PKS01}$, for (a) $\epsilon$=100 cm$^2$/s$^3$ and (b) $\epsilon$=400 cm$^2$/s$^3$.

(FDM), to maintain the kinetic energy with $|\mathbf{k}| < 2.5$, where $\mathbf{k}$ is a wavevector. Spatial derivatives are calculated using fourth-order central differences. The conservative scheme of Morinishi et al. (1998) is employed for the advection term, and the second-order Runge-Kutta scheme is employed for time integration. To solve the velocity-pressure coupling, we use the highly simplified marker and cell (HSMAC) scheme (Hirt and Cook (1972)), which iterates until the rms of the velocity divergence becomes smaller than $\delta/\Delta$, where $\Delta$ is the grid spacing and $\delta$ is chosen to be $10^{-3}$. The governing equations are discretized by using a cubic domain of length $2\pi L_0$, where $L_0$ is the representative length. Periodic boundary conditions are applied in all three directions. The flow cube is discretized uniformly into $N^3$ gridpoints, resulting in $\Delta = 2\pi L_0/N$.

Under the limit of a large ratio of the density of the particle material to that of the fluid ($\rho_p/\rho_f >> 1$), the governing equation for water droplets is given by

$$\frac{d\mathbf{V}}{dt} = -\frac{f}{\tau_p}\left(\mathbf{V} - (\mathbf{U}(\mathbf{x},t) + \mathbf{u}(\mathbf{x},t))\right) + \mathbf{F}_{impulse} + \mathbf{g}, \qquad (26)$$

where $\mathbf{V}$ is the particle velocity, $\mathbf{U}$ is the air velocity at the position of the droplet, $\mathbf{u}$ is the disturbance flow velocity due to the surrounding droplets, and $\tau_p$ is the particle relaxation time defined as $\tau_p = (2/9)(\rho_p/\rho_f)(r^2/\nu)$, in which $r$ is the particle radius. $\mathbf{F}_{impulse}$ denotes the impulsive force due to collisions and $\mathbf{g}$ is the gravity vector ($= (-g,0,0)$), where $g$ is the gravitational acceleration). The ratio of the density of the particle material to that of the fluid, $\rho_p/\rho_f$, is set to $8.43\times10^2$ at 1 atm and 298 K, and $f$ is the drag coefficient defined as the ratio between the nonlinear drag and the linear drag (Rowe and Henwood (1961)).

The second-order Runge-Kutta method is used for the time integration. The flow velocity at the droplet position $\mathbf{U}$ is linearly interpolated from the adjacent grid values. This simple linear interpolation is justified through comparisons with the cubic





|  | $N^3$ | $L_0$ [m] | $Re$ | $u'$ | $k_{max}l_\eta$ | $Re_\lambda$ | $N_p$ |
|---|---|---|---|---|---|---|---|
| N4000 | $4000^3$ | 0.312 | 14100 | 1.01 | 2.10 | 874 | $1.60 \times 10^9$ |
| N6000 | $6000^3$ | 0.468 | 24200 | 1.01 | 2.11 | 1140 | $5.40 \times 10^9$ |

**Table 3.** Case configurations and typical turbulence statistics. $Re = U_0 L_0/\nu$, $u'$ is the rms of flow velocity fluctuation, $k_{max} (= N/2)$ is the maximum wavenumber, $l_\eta$ is the Kolmogorov scale, and $Re_\lambda$ is the Taylor-microscale based Reynolds number. $N_p$ is the total number of particles.

Hermitian, cubic Lagrangian, and fifth-order Lagrangian interpolations from Sundaram and Collins (1996). The disturbance flow $\mathbf{u}$, which denotes the hydrodynamic interaction, is calculated by using the BiSM (Onishi et al. (2013)). The particle mass and volume fractions are so dilute that the flow modulation is ignored.

### 3.2 *Computation for turbulent collision statistics*

After the background airflow has reached a statistically stationary state, monodispersed water droplets are introduced into the flow. After a period exceeding three times the eddy-turnover time $T_0 = L_0/U_0$, collision detection is then started. Droplets are allowed to overlap (ghost-particle condition) and a collision is judged from the trajectories of a pair of droplets by assuming linear particle movement for the time interval $\Delta t$.

The detailed description of the procedures for calculating collision statistics can be found in Onishi et al. (2013), who conducted the DNS for $Re_\lambda$ up to 530. This study performed additional simulations to push the maximum $Re_\lambda$ forward, up to 1,140. The computational settings for the present simulations are summarized in Table 3.

### 3.3 Computation for size evolutions due to collisional growth

To obtain reference data regarding droplet collisional growth, we tracked the growth of droplets that initially had the following exponential size distribution (e.g., Soong (1974)):

$$f_0(x) = \frac{n_0}{x_{m0}} \exp(-x/x_{m0}), \tag{27}$$

where $x_{m0}$ is the mass of a droplet with a radius of $r_{m0}$ and $n_0$ is the initial number density. We carried out two cases: one with $r_{m0}$=15 $\mu$m and $n_0 = 1.42 \times 10^8$ m$^{-3}$, and the other with $r_{m0}$=10 $\mu$m and $n_0 = 4.79 \times 10^8$ m$^{-3}$. The corresponding initial liquid water content was 2.0 g/m$^3$ for both cases. It was assumed that colliding particles immediately united without breakups, and conserved mass and momentum.

Table 4 summarizes the computational parameters for the flow calculation as well as the obtained flow statistics for the collision growth simulations. In cases T100, T, and T1000, the same grid configuration with the same Reynolds number was calculated, but the energy dissipation rates, which are in the typical range observed in turbulent atmospheric clouds, were 100, 400, and 1,000 cm$^3$/s$^2$, respectively. Cases T, TR127, TR206, and TR333 obtained flows with the same energy dissipation





|  | $N^3$ | $L_0$ [m] | $Re$ | $u'$ | $k_{max}l_\eta$ | $Re_\lambda$ | $\epsilon$ [cm$^3$/s$^2$] |
|---|---|---|---|---|---|---|---|
| NoT | $32^3$ | 0.0127 | 0 | 0 | - | 0 | 0 |
| T100 | $96^3$ | 0.0180 | 97.4 | 1.00 | 2.04 | 66.1 | 100 |
| T | $96^3$ | 0.0127 | 97.4 | 1.00 | 2.04 | 66.1 | 400 |
| T1000 | $96^3$ | 0.0101 | 97.4 | 1.00 | 2.04 | 66.1 | 1000 |
| TR127 | $256^3$ | 0.0338 | 360 | 0.98 | 2.06 | 127 | 400 |
| TR206 | $512^3$ | 0.0669 | 908 | 1.00 | 2.06 | 206 | 400 |
| TR333 | $1000^3$ | 0.135 | 2220 | 1.00 | 2.07 | 333 | 400 |

**Table 4.** Case configurations and typical turbulence statistics. $Re = U_0 L_0/\nu$, where $U_0$ is the representative velocity and $L_0$ is the representative length, $u'$ is the rms of the flow velocity fluctuation, $k_{max}(= N/2)$ is the maximum wavenumber, $l_\eta$ is the Kolmogorov scale, $\lambda$ is the local shear rate, and $Re_\lambda$ is the Taylor-microscale-based Reynolds number.

rate (400 cm$^3$/s$^2$) but with different $Re_\lambda$ values. Onishi et al. (2015) already presented these cases except for TR333 with $r_{m0}$=15 $\mu$m. The present study additionally performed the case TR333 with $r_{m0}$=15 $\mu$m to obtain a clear Reynolds-number dependence, as well as cases T, TR127, and TR206 with $r_{m0}$=10 $\mu$m.

## 4 Results and Discussion

### 4.1 Estimate for Reynolds-number dependence of clustering effect of small-$St$ particles

Onishi et al. (2013) observed that the clustering effect and consequently the collision kernel decreases as the Reynolds number increases for $Re_\lambda$>100 and $St$=0.4. Later, Onishi and Vassilicos (2014) clarified that the Reynolds-number dependence of $g_{11}$ observed for $1/3 < St < 1$ is due to internal intermittency of the three-dimensional turbulence.

To quantify the influence of intermittence on $g_{11}$, we need to separate the local quantity from the global (average) quantity. Kolmogorov (1962) introduced the local energy dissipation as

$$\epsilon_l(\mathbf{x},t) = \frac{3}{4\pi l^3} \int_{|\mathbf{y}| \leq l} \epsilon^\#(\mathbf{x}+\mathbf{y},t)d\mathbf{y}, \tag{28}$$

where superscript # denotes the local quantity. It was supposed that the PDF of $\epsilon_l$ follows a log-normal distribution if $l$ is much smaller than the flow integral scale. Assuming $l \sim \eta$, we obtain

$$P_{LN}(\epsilon^*|\mu,\sigma^2) = \frac{1}{\sqrt{2\pi}\sigma\epsilon^*} \exp\left(\frac{-(\ln\epsilon^* - \mu)^2}{2\sigma^2}\right), \tag{29}$$

where $\epsilon^* = \epsilon_\eta$. Parameters $\sigma$ and $\mu$ appear in the first and second moments of $\epsilon^*$ as

$$\langle \epsilon^* \rangle (= \epsilon) = \exp\left(\mu + \sigma^2/2\right) \tag{30}$$





and

$$\langle \epsilon^{*2} \rangle = \exp\left(2\mu + 2\sigma^2\right), \tag{31}$$

respectively.

The intermittency is measured by the flatness factor $F$, defined as

$$F = \frac{\left\langle (\partial u_1/\partial x_1)^4 \right\rangle}{\left\langle (\partial u_1/\partial x_1)^2 \right\rangle^2}. \tag{32}$$

It is observed that $F$ follows a power law relation with $Re_\lambda$, for example, $F \sim Re_\lambda^{3/8}$ (Pope (2000)). Given $\partial u_1/\partial x_1 \sim (\epsilon_\eta/\nu)^{1/2} = (\epsilon^*/\nu)^{1/2}$, we obtain

$$F \sim \frac{\langle \epsilon^{*2} \rangle}{\epsilon^2} \sim Re_\lambda^{3/8}. \tag{33}$$

Substitution of Eqs. (30) and (31) into Eq. (33) yields

$$\sigma^2 = \frac{3}{8}\ln\left(Re_\lambda\right). \tag{34}$$

Eq. (30) then yields

$$\mu = \ln \epsilon Re_\lambda^{-3/16}. \tag{35}$$

That is, $P_{LN}(\epsilon^*|\mu,\sigma^2)$ can be rewritten as $P_{LN}(\epsilon^*|Re_\lambda)$.

We can define a local $St$, $St^*$, as

$$St^* = St \times \left(\frac{\epsilon^*}{\epsilon}\right)^{1/2}, \tag{36}$$

the PDF of which follows

$$P(St^*|Re_\lambda) = \frac{2\epsilon St^*}{St^2} P_{LN}\left(\epsilon\left(\frac{St^*}{St}\right)^2 \middle| Re_\lambda\right). \tag{37}$$

It should be emphasized that the shape of $P_{LN}$ (and consequently $P$) depends on $Re_\lambda$. If we assume a universal radial distribution function at contact separation against $St^*$ $-g_{11}^{\#univ}(St^*)-$, the global clustering effect can be obtained as

$$g_{11}(St, Re_\lambda) = \int_0^\infty g_{11}^{\#univ}(St^*)P(St^*|Re_\lambda)dSt^*. \tag{38}$$

It should be noted that $g_{11}$ depends on $Re_\lambda$, whereas $g_{11}^{\#univ}$ does not (which is why it is called *universal*). For $St^* \ll 1$, the universal clustering effect would have the form $g_{11}^{\#univ} = A_1 St^{*2} + 1$ by following Eq. (10). Substitution of this form into Eq.





(38) yields $g_{11}(St \ll 1, Re_\lambda) = A_1 St^2 + 1$, regardless of the value of $Re_\lambda$. This explains why the $g_{11}$ for $St = 0.1$ does not show a significant Reynolds-number dependence. For a moderate $St^*$, we simply formulate the universal function by following Eqs. (10) and (11) but without the smoothing operators, as follows:

$$g_{11}^{\#univ}(St^*) = H(St^* - St_a^*)A_1^* St^{*2} + H(St_a^* - St^*)A_2^* St^{*-2}, \tag{39}$$

where $A_1^*$ and $A_2^*$ are empirical parameters and $St_a^*$ is defined as $(A_2^*/A_1^*)^{1/4}$. Based on the DNS data for $St$=0.1, 0.4, and 0.6 in the flow with $Re_\lambda$=130, we found that $A_1^* = 110$ and $A_2^* = 0.073$ work reasonably well. Although we have no justification for this universal function, it can provide $g_{11}$ for arbitrary $St$ (<1) and $Re_\lambda$ through Eq. (38). We calculated $g_{11}$ for $St$=0.1, 0.4, and 0.6 with $Re_\lambda$= 100, 200, 400, 1,000, 4,000 and 10,000. We then obtained the following empirical formulations by applying the least squre method to the calculated results.

$$g_{11}(St = 0.1, Re_\lambda) \quad \sim \quad 2.1, \tag{40}$$

$$g_{11}(St = 0.4, Re_\lambda) \quad \sim \quad 19.3 - 1.9 \log_{10} Re_\lambda, \tag{41}$$

$$g_{11}(St = 0.6, Re_\lambda) \quad \sim \quad 34.3 - 3.9 \log_{10} Re_\lambda. \tag{42}$$

Figure 3 shows a comparison between $g_{11}$ values from the above equations and those from the DNS. The figure shows that the empirical estimates can reproduce the Reynolds-number dependence of $g_{11}$ correctly.

## 4.2 Modeling of clustering effect

Figure 3 shows a comparison between direct numerical simulation results and model predictions for $g_{11}$. The dashed lines are the prediction by the Onishi model (Onishi et al. (2015)), and the solid lines are the predictions by the present updated model. The DNS data for $St \leq 1$ and for $Re_\lambda \leq 530$ were obtained from the table in Onishi et al. (2015). The data for $St$=1.4, 2, 4, and 8 were newly obtained. The results for $Re_\lambda$=874 and 1,140 (these Reynolds numbers are the largest ever achieved for turbulent particle collision statistics) are included in the figure. The DNS data show a decreasing trend for $St < 1$ for the moderate Reynolds number range of $100 \lesssim Re_\lambda \lesssim 1000$. This decreasing trend with respect to $Re_\lambda$ is attributed to the flow intermittency (Onishi and Vassilicos (2014)) as discussed in the previous subsection. The black solid line is the estimated $g_{11}$ for $St = 0.4$ and the black dashed line is for $St = 0.6$ (Eqs. (41) and (42), respectively). The present Onishi model show slightly better agreement with the DNS data in terms of the slopes in comparison with the original model. For $St > 1$, the DNS data show increasing trends for the moderate $Re_\lambda$ range, and those trends are predicted by the present parameterization, although the rate for $St = 2$ is overestimated. One significant feature of the Onishi $g_{11}$ model is that maximum clustering occurs at a larger $St$ for a larger $Re_\lambda$. This shows a clear contrast with the Ayala-Wang model, which was designed to show maximum clustering at $St \sim 1$ regardless of $Re_\lambda$.

The updated parameterization leads to improvement, particularly for the $St \geq 1$ regime. For example, in the case of $Re_\lambda = 127$, the rms values of the relative errors of the prediction with the original parameters for (i) $St$=0.1, 0.2, 0.4, and 0.6 and for (ii) $St$=1, 1.4, 2, 4, and 8 were (i) 0.081 and (ii) 0.239. The rms values with the present parameters were (i) 0.075 and (ii) 0.113.



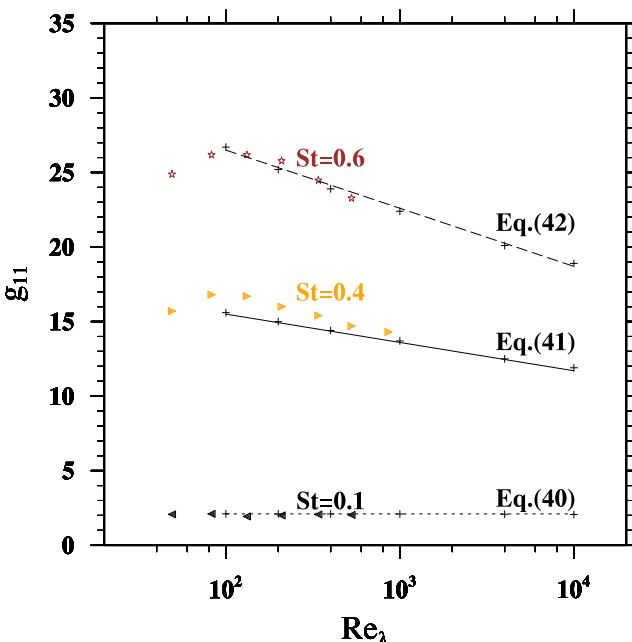

**Figure 2.** Radial distribution function at the contact of monodisperse particles with $St$=0.1, 0.4, and 0.6 against $Re_\lambda$. The plotted symbols are the reference DNS results. The lines are the results of Eqs. (40), (41), and (42), which were fitted to the sample values (+) with using the least square method.

### 4.3 Turbulent coagulation kernels for small Reynolds-number flow

Figure 4 shows a comparison between model predictions and DNS results of the coagulation kernel $K_{coag}(r_1, r_2)$ for $r_1$=30 $\mu$m, $Re_\lambda$=127 and $\epsilon$=400 cm²/s³. The kernel is normalized by the collision radius $R$ and the local velocity gradient $\lambda \left( = (\epsilon/\nu)^{1/2} \right)$. The reference DNS considers the hydrodynamic interaction and the gravitational droplet sedimentations. We observe a large

5    discrepancy for $r_2 \sim 30$ $\mu$m (=$r_1$), where the turbulence enhancement on collision efficiency is difficult to define, because the collision efficiency for $r_1 = r_2$ cannot be defined for stagnant flow. Otherwise, the model predictions (Ayala-Wang model and Onishi model) agree well with the DNS results. As an example, we also observe a slight improvement of the Onishi model by including the sedimentation effect on $\langle |w_r| \rangle$ (subsection 2.3.2) on the data for $r_2 = 40$ $\mu$m.

The Ayala-Wang model shows a local maximum around $r_2 = r_1$. The DNS results also show a convex shape, but the value at

10   $r_2 = r_1$ is much smaller than the prediction by the Ayala-Wang model. In contrast, the Onishi model does not show such a local





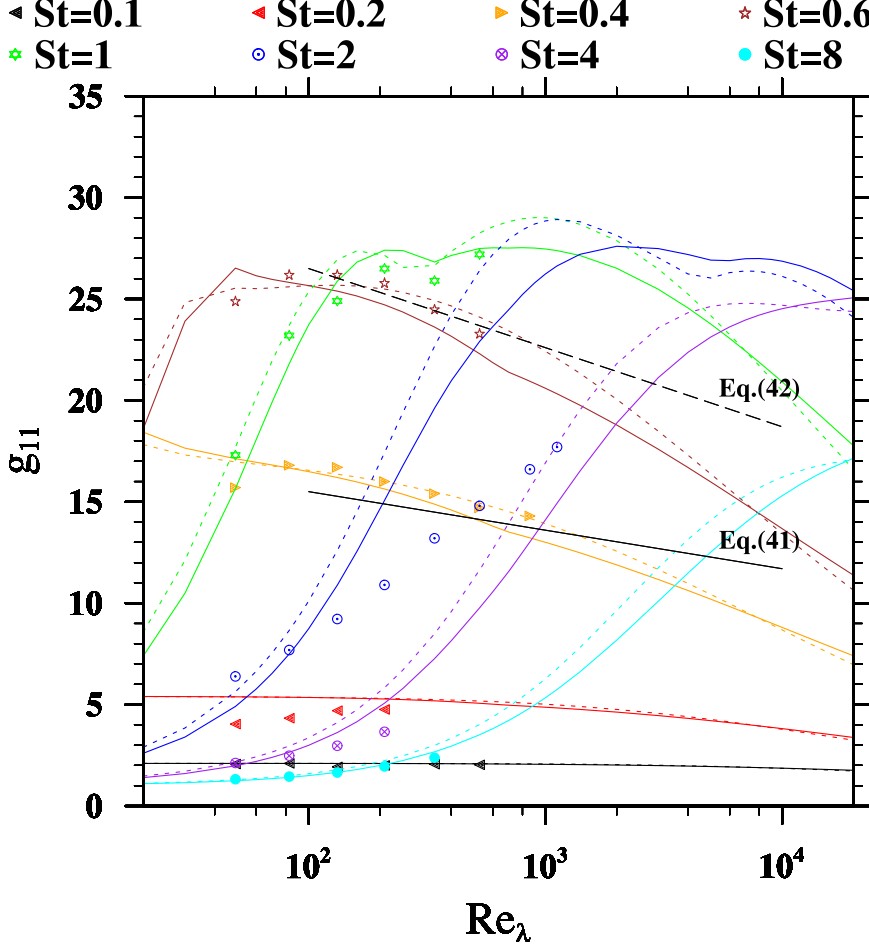

**Figure 3.** Radial distribution function at the contact of monodisperse particles, $g_{11}$, against $Re_\lambda$. The plotted points are the reference DNS results, the dotted lines are the prediction with the coefficients of Onishi et al. (2015), and the solid lines are the present prediction.

maximum at $r_2 = r_1$ but does provide values much closer to DNS elsewhere. The convex shape is related to the diffusion effect denoted by $r_d$ in Eq. (7). Eq. (16) for $g_{12}$, employed in the Onishi model, was formulated for non-sedimenting droplets and this equation therefore leads to weaker acceleration-driven diffusion, i.e., smaller $r_d$ (Ayala et al. (2008a)). This can explain why the Onishi model does not show the convex shape.

5    Figure 5 shows the ratio of the turbulent coagulation kernel to the Hall kernel for the turbulent flow with $Re_\lambda$=127 and $\epsilon$=400 cm$^2$/s$^3$. The level of the ratio is basically similar for both the Ayala-Wang and Onishi models, and the ratio is nearly unity when the droplets are above 100 $\mu$m.





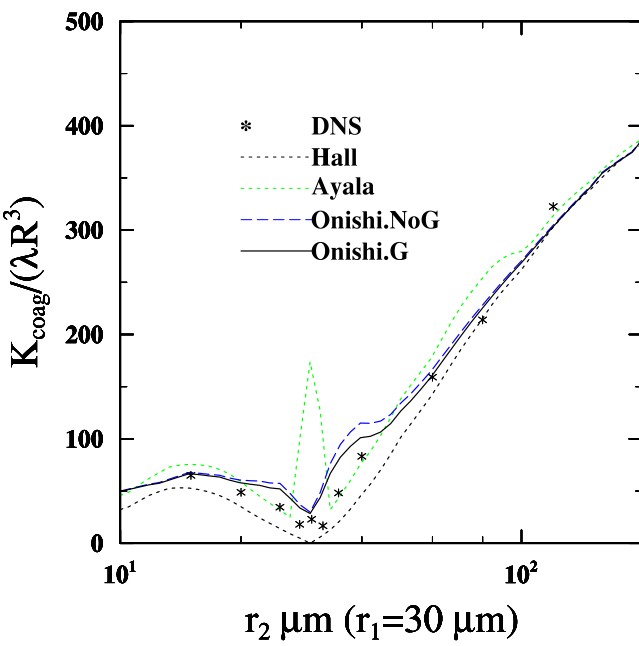

**Figure 4.** Non-dimensionalized coagulation kernels for $r_1$=30 $\mu$m in the turbulent flow with $Re_\lambda$=127 and $\epsilon$=400 cm$^2$/s$^3$.

### 4.4 Reynolds-number dependence of kernel models

Figure 6 shows the ratio of the coagulation kernel for $Re_\lambda$=10$^4$ to that for $Re_\lambda$=10$^3$. It should be noted that the $E_c$ and $\eta_E$ models employed in the Ayala-Wang and Onishi kernels do not consider the Reynolds dependence. Therefore, the figure actually shows the ratio of the geometric collision kernels, i.e., the ratio of $|w_r|g_{12}$. The Ayala-Wang kernel increases for the autoconversion region ($r_1, r_2 < 40$ $\mu$m) and the accretion region ($r_1 < 40$ $\mu$m and $r_2$>40 $\mu$m, and $r_1 > 40$ $\mu$m and $r_2$<40 $\mu$m). The Onishi kernel decreases for the corresponding autoconversion region, but increases for the rain-rain self-collection region ($r_1, r_2 > 40$ $\mu$m).

Figure 7 shows the ratio of $g_{12}$ for $Re_\lambda$=10$^4$ to that for $Re_\lambda$=10$^3$. It should be noted that the form of Eq. (22) violates the spherical form and we cannot rigorously define $g_{12,total}$ and $\langle|w_r|\rangle_{total}$ that formulate $K_{c,total} = 2\pi R_{12}^2 \langle|w_r|\rangle_{total}\, g_{12,total}$. Here, we simply considered $g_{12}$ expressed by Eq. (11) as the $g_{12,total}$ for the total kernel and obtained $\langle|w_r|\rangle_{total} = K_{c,total}/\left(2\pi R^2 g_{12}\right)$. As designed, the Ayala-Wang kernel shows the increase for increasing $Re_\lambda$ for both the autoconversion and the accretion regions. In contrast, the Onishi kernel shows a decrease for the autoconversion region, but a significant increase for the accretion




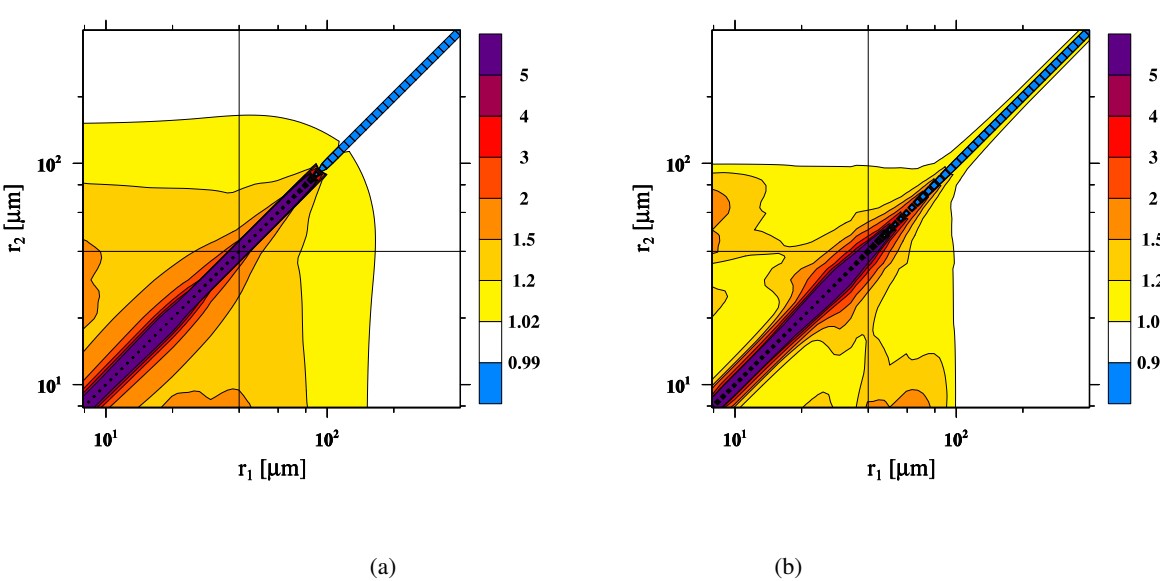

(a)                                             (b)

**Figure 5.** Ratio of the turbulent coagulation kernel to the Hall kernel in the turbulent flow with $Re_\lambda$=127 and $\epsilon$=400 cm$^2$/s$^3$. (a) Ayala-Wang kernel and (b) the present Onishi kernel.

region and the rain-rain self-collection region (i.e., $r_1, r_2 > 40$ $\mu$m). This is due to the shift of the maximum clustering toward larger $St$ with increasing $Re_\lambda$.

Figure 8 shows the ratio of the radial relative velocity for $Re_\lambda$=10$^4$ to that for $Re_\lambda$=10$^3$. The Ayala-Wang kernel shows little Reynolds-number dependence. In contrast, the Onishi kernel shows significant Reynolds-number dependence, which tends to
5    oppose the Reynolds-number dependence of $g_{12}$ and thus weakens the Reynolds-number dependence of the collision kernel.

The Reynolds-number dependence of the clustering effect is larger than that of the radial relative velocity, and the contour shape of Figure 6 is more similar to Figure 7 than to Figure 8 for both the Ayala-Wang and the Onishi kernels. That is, the Reynolds-number dependence of the two kernels can mostly be attributed to the $g_{12}$ parameterizations.

Note that the Fortran 90 code used to calculate the present Onishi kernel is provided as a supplemental material.

10   **4.5   Turbulence enhancement of autoconversion rate**

We investigated the turbulence enhancement on the autoconversion rate, which is the conversion rate from the cloud category ($r$ <40 $\mu$m) to the rain category due to collisions between the small cloud droplets. The Ayala-Wang kernel and the present Onishi kernel were employed to calculate the coagulation growth of droplets modeled by the stochastic collision-coalescence equation (SCE):





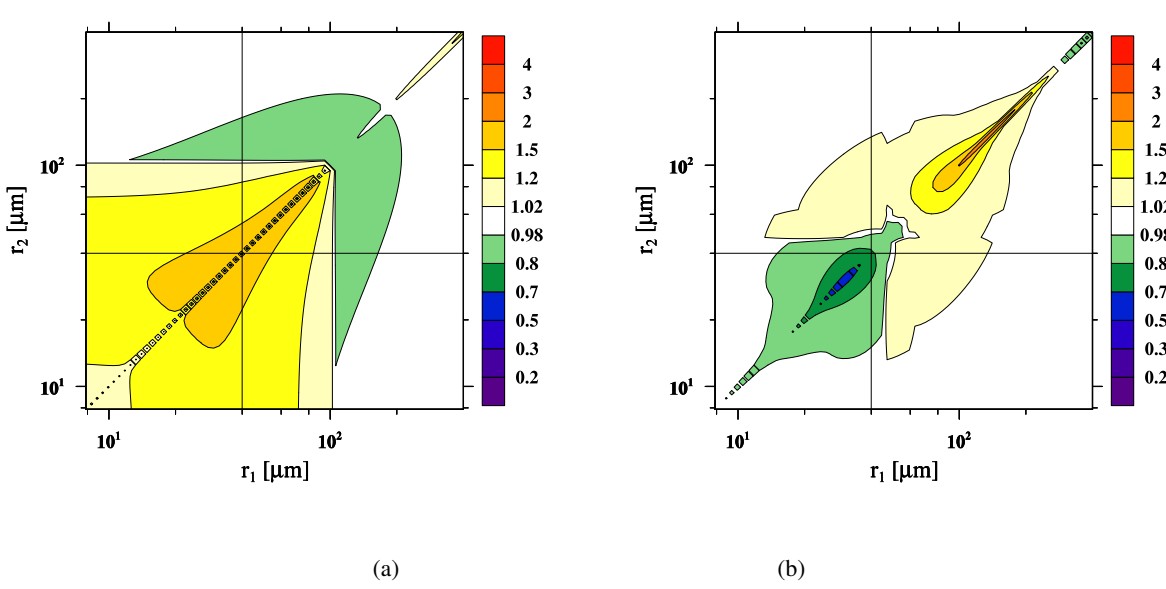

(a)  (b)

**Figure 6.** Ratio of the coagulation kernel for $Re_\lambda=10^4$ to that for $Re_\lambda=10^3$. (a) Ayala-Wang kernel and (b) the present Onishi kernel.

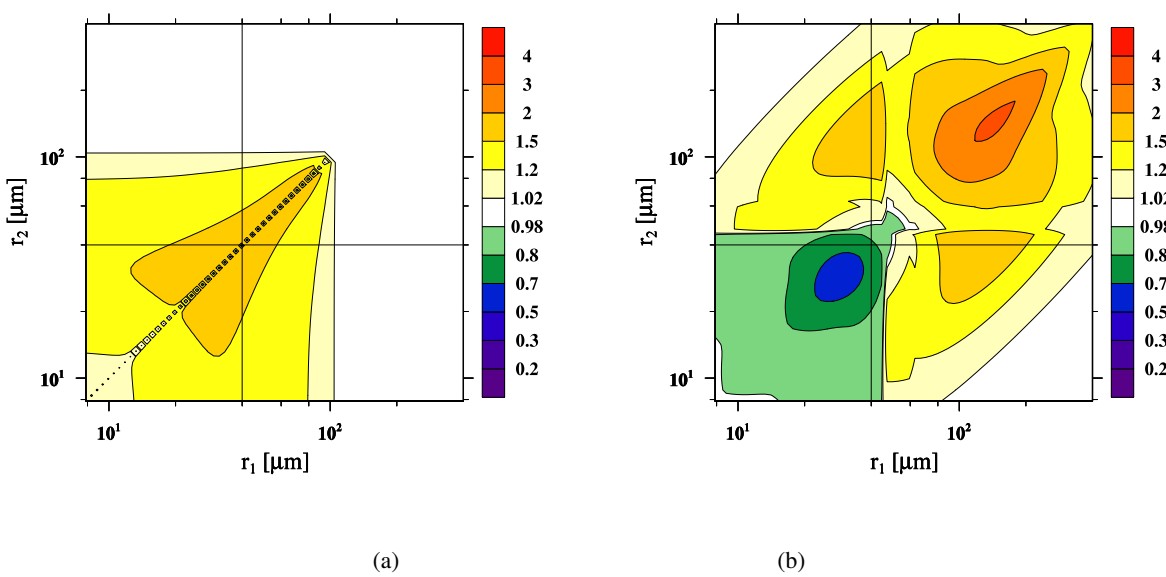

(a)  (b)

**Figure 7.** Ratio of the clustering effect $g_{12}$ for $Re_\lambda=10^4$ to that for $Re_\lambda=10^3$.





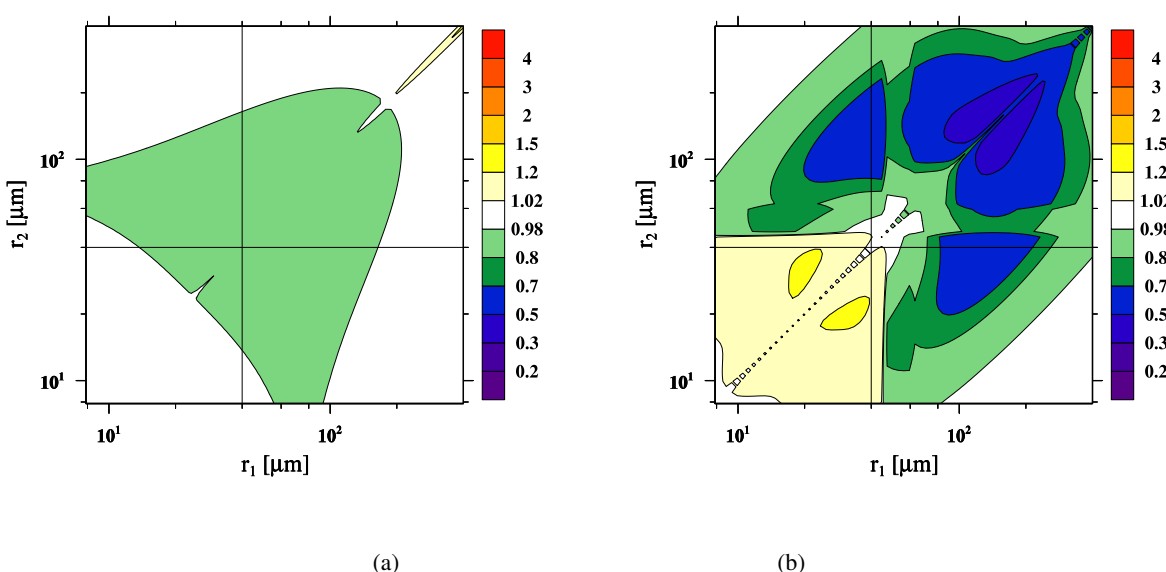

(a)                                    (b)

**Figure 8.** Ratio of the radial relative velocity at contact separation $\langle |W_r| \rangle$ for $Re_\lambda = 10^4$ to that for $Re_\lambda = 10^3$.

$$\frac{\partial n_f(m,t)}{\partial t} = \int_0^{m/2} K_{coag}(m-m',m')n_f(m-m',t)n_f(m,t)dm'$$

$$- \int_0^{\infty} K_{coag}(m,m')n_f(m,t)n_f(m',t)dm', \tag{43}$$

where $m$ is the particle mass and $n_f$ is the number density function. The coagulation component of the spectral bin model in the Multi-Scale Simulator for the Geoenvironment (MSSG-Bin) cloud physics model (Onishi and Takahashi (2012)) was used to solve the SCE. The mass coordinate $m$ was discretized as $m_k = 2^{1/s}m_{k-1}$, where $s$ was set to 16. The representative radius of the first bin was 2.7 $\mu$m and 528 classes were calculated, the largest class of which had a representative radius of 5.4 mm. The SCE solution is basically a mean-field approximation. In contrast, the LCS acts as a reference model as it includes all turbulence effects directly in its Lagrangian particle simulation. Due to the high computational cost, however, the LCS is restricted to moderate Reynolds number (here up to $Re_\lambda = 333$).

Following Seifert et al. (2010), Onishi et al. (2015) used a quantitative measure of the turbulence enhancement focusing on the timescale of the autoconversion process. The time required for a cloud to convert 10% of its cloud mass into rain category



drops is expressed as $t_{10\%}$, which can be used as a measure of the autoconversion timescale. Then, we can define the turbulence enhancement factor, $E_{turb}$, as

$$E_{turb} = \frac{P_{auto}|_T}{P_{auto}|_{NoT}} = \frac{\overline{t_{10\%}}_{NoT}}{\overline{t_{10\%}}_T}, \tag{44}$$

where the overbar indicates the mean value.

Figure 9(a) shows $E_{turb}$ as a function of $\epsilon$ for $Re_\lambda$=66 in the $r_{m0}$=10 $\mu$m case. The LCS data show an almost linear increase with increasing $\epsilon$. Both the SCE simulation with the Ayala-Wang kernel (SCE-Ayala hereafter) and that with the Onishi kernel (SCE-Onishi hereafter) show the same trend with the LCS data, although the SCE-Ayala slightly overestimates the enhancement. The maximum relative difference between the SCE-Ayala and SCE-Onishi kernels was as small as 22% at $\epsilon$=500 cm$^2$/s$^3$. Both the SCE-Ayala and the SCE-Onishi kernels show a kink at $\epsilon$=600 cm$^2$/s$^3$, where the turbulence enhancement on collision efficiency levels off. Figure 9(b) shows $E_{turb}$ as a function of $Re_\lambda$ for $\epsilon$=400 cm$^2$/s$^3$ in the case of $r_{m0}$=10 $\mu$m. The SCE-Ayala and the SCE-Onishi kernels show different trends: the SCE-Ayala predicts an increasing enhancement with increasing $Re_\lambda$, while the SCE-Onishi predicts almost constant or slightly decreasing enhancement. The difference between the two SCE predictions becomes larger for larger $Re_\lambda$, with the LCS result closer to the SCE-Onishi prediction. The difference between the SCE-Ayala and the SCE-Onishi kernels can be explained by the Reynolds-number dependence of the two kernels, as discussed in subsection 4.4. This Reynolds-number dependence is relevant, because the SCE prediction becomes very different at large $Re_\lambda$. For example, at $Re_\lambda = 2 \times 10^4$, the SCE-Ayala prediction is 2.5 times larger than the SCE-Onishi prediction. The LCS results for $Re_\lambda \leq 206$ support the SCE-Onishi prediction.

Figure 10 shows $E_{turb}$ for the $r_{m0}$=15 $\mu$m case, which was also discussed in Onishi et al. (2015). This study additionally performed the simulation for $Re_\lambda$=333 to investigate the Reynolds-number dependence more clearly. Basically, the results are similar to those in the previous figure. In Figure 10, the SCE-Ayala and the SCE-Onishi kernels show closer results for $Re_\lambda$=66, and both SCE-Ayala and SCE-Onishi slightly overestimate the enhancement for $\epsilon$>400 cm$^2$/s$^3$. The difference between the two predictions at $Re_\lambda = 2 \times 10^4$ is larger: the SCE-Ayala prediction is 3.0 times larger than the SCE-Onishi prediction. The LCS results for $Re_\lambda$ up to 333 clearly support the SCE-Onishi prediction.

In summary, both Figures 9 and 10 show that the SCE-Ayala and the SCE-Onishi kernels produce consistent results for low $Re_\lambda$ with about a 20% difference at most, but the two show very different values at large $Re_\lambda$: the SCE-Ayala prediction becomes larger than the SCE-Onishi by a factor of up to 3 in cloud turbulence. This clearly suggests a strong demand for collision growth data with larger $Re_\lambda$ to construct a more robust turbulent kernel.

## 5 Conclusions

This study investigated the Reynolds-number dependence of turbulence enhancement on the collision growth of cloud droplets. The Onishi turbulent coagulation kernel proposed in Onishi et al. (2015) was updated by using the present direct numerical simulation (DNS) results for the Taylor-microscale-based Reynolds number ($Re_\lambda$) up to 1,140. The following three components





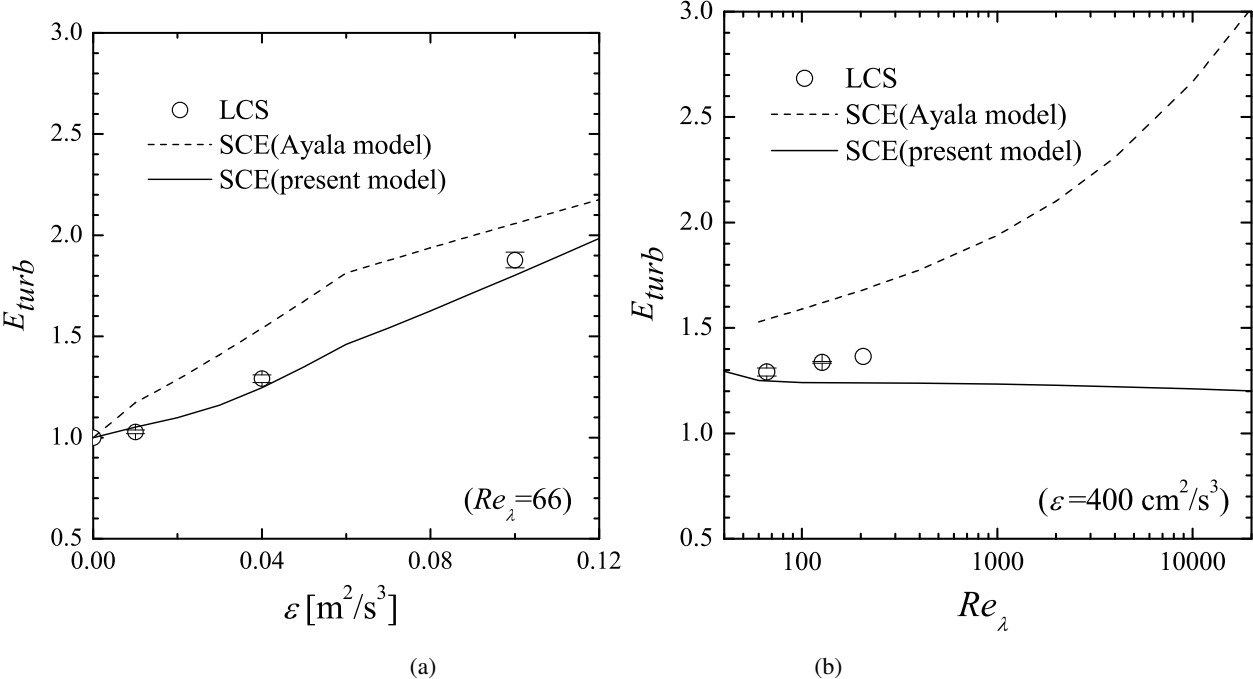

**Figure 9.** Turbulence enhancement factors for $\overline{r_c}$=10 $\mu$m as a function of (a) the energy dissipation rate $\epsilon$ and (b) the Taylor-microscale-based Reynolds number $Re_\lambda$. $Re_\lambda$=66 in (a) and $\epsilon$=400 cm$^2$/s$^3$ in (b). The error bars indicate the standard deviations.

were updated: (i) the radial distribution function at contact separation of a monodisperse suspension of droplets, i.e., the clustering effect, $g_{11}$, (ii) the radial relative velocity at contact separation, $\langle|w_r|\rangle$, and (iii) the turbulence enhancement on collision efficiency, $\eta_E$.

We confirmed that the updated $g_{11}$ parameterization agrees better with DNS results than the original parameterization for

$Re_\lambda \sim 100$. We also confirmed that the updated parameterization has better agreement with the Reynolds-number dependence of $g_{11}$ for the estimated values of $St = 0.4$ and 0.6. The model of radial relative velocity was updated to include the effect of the gravitational sedimentation of droplets. The comparison with the DNS results confirmed that the updated model for $\langle|w_r|\rangle$ is better than the original one. The Onishi coagulation kernel employed the turbulence enhancement on collision efficiency $\eta_E$, tabulated in Wang et al. (2008). The updated kernel is intended to adjust to more recent $\eta_E$ values, tabulated in Wang

and Grabowski (2009). It should be noted that the collision efficiency $E_c$ in Pinsky et al. (2001) ($E_{c,PKS01}$), which the Onishi kernel employs, is different from the $E_c$ in Hall (1980) ($E_{c,Hall}$), particularly for $r_2/r_1 \sim 0$ or $\sim 1$. We proposed a compensation such that $\eta_E$ (in Wang and Grabowski (2009)), which shows the turbulence enhancement against $E_{c,Hall}$, is applicable to the kernel with $E_{c,PKS01}$. The proposed compensation is simply to multiply $\eta_E$ in Wang and Grabowski (2009) by $E_{c,PKS01}/E_{c,Hall}$.

The present Onishi coagulation kernel was compared with the Ayala-Wang kernel (Ayala et al. (2008a); Wang et al. (2008)) together with the DNS values for $Re_\lambda$=66 and the energy dissipation rate $\epsilon$=400 cm$^2$/s$^3$. For $K_{coag}(r_1 = 30\mu$m$, r_2)$, both





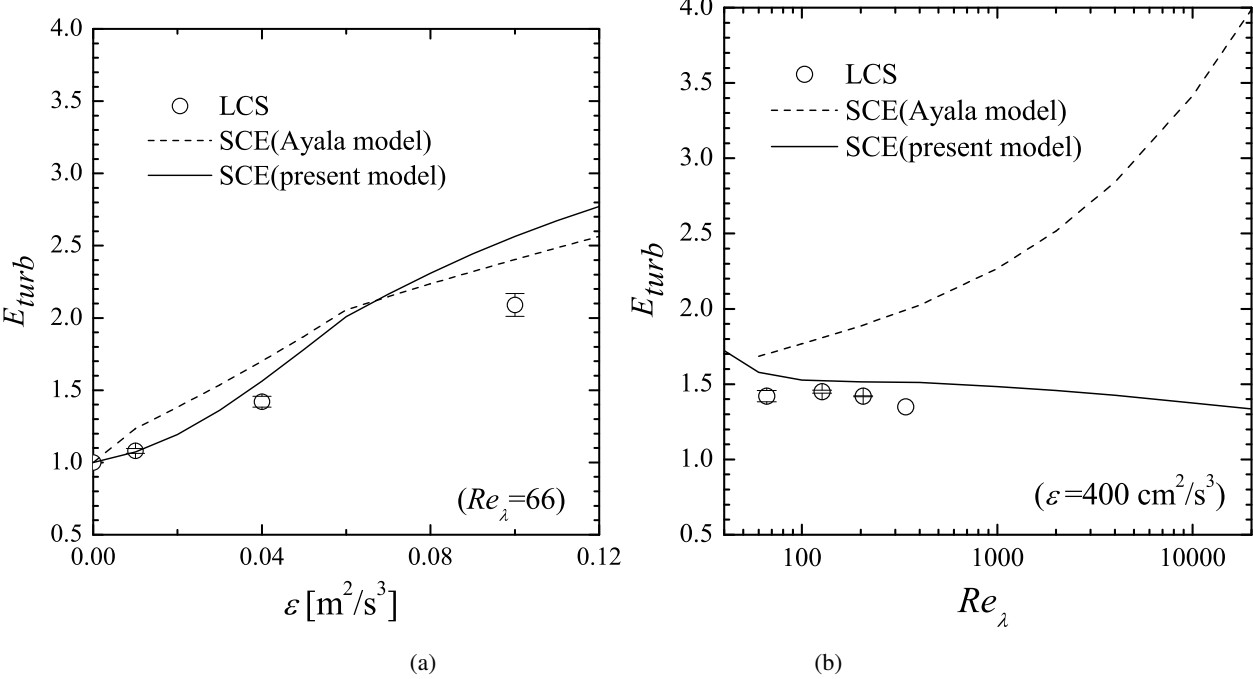

(a)                                        (b)

**Figure 10.** Turbulence enhancement factors for $\overline{r_c}$=15 as a function of (a) the energy dissipation rate and (b) the Taylor-microscale- based Reynolds number $Re_\lambda$. $Re_\lambda$=66 in (a) and $\epsilon$=400 cm$^2$/s$^3$ in (b). The error bars indicate the standard deviations.

kernels show similar values comparable to the DNS values except for $r_2 \sim r_1$. For the nearly monodisperse case, the Ayala-Wang kernel overestimates the kernel but provides a sharp convex shape, i.e., a clear local maximum at $r_2 = 30 \ \mu$m, that agrees with the DNS data qualitatively. The Onishi kernel does not show such a convex shape due to weaker acceleration-driven diffusion on the clustering effect $g_{12}$, but the kernel values are in fairly good agreement with the DNS. The Reynolds-number dependence of the two kernels was also compared. It was shown that the Ayala-Wang kernel increases for the autoconversion region ($r_1, r_2 < 40 \ \mu$m) and the accretion region ($r_1 < 40 \ \mu$m and $r_2$>40 $\mu$m, and $r_1 > 40 \ \mu$m and $r_2$<40 $\mu$m). In contrast, the Onishi kernel decreases for the autoconversion region but increases for the rain-rain self-collection region ($r_1, r_2 > 40 \ \mu$m). These Reynolds-number dependences can be attributed to the Reynolds-number dependence of the clustering effect.

We also compared the stochastic collision-coalescence equation (SCE) simulations for both kernels; one with the Ayala-Wang kernel (SCE-Ayala) and the other with the present Onishi kernel (SCE-Onishi). Lagrangian Cloud Simulator (LCS, Onishi et al. (2015)) simulations were also conducted to obtain reference data of the turbulent enhancement on collisional growth, in particular, the enhancement on the autoconversion rate. The SCE-Ayala and SCE-Onishi kernels show consistent results for $Re_\lambda$=66 with about a 20% difference at most, but the two SCE simulations show a different Reynolds-number dependence, resulting in large differences at large $Re_\lambda$. It should be emphasized that the SCE-Ayala prediction can become larger than the SCE-Onishi by a factor of up to 3 in the typical large $Re_\lambda$ range observed in cloud turbulence. These simulations





clearly suggest a strong demand for reference collision growth data with larger $Re_\lambda$ from DNS or laboratory measurement to construct a more robust kernel model. This is our goal in future studies.

*Acknowledgements.* Part of the presented simulations were performed on the supercomputer Earth Simulator at the Japan Agency for Marine-Earth Science and Technology. The large-size simulations for collision statistics for $Re_\lambda$=874 and 1,140 were performed on the K-computer

5   provided by the RIKEN Advanced Institute for Computational Science through the HPCI System Research project (Project ID: hp140120). We thank Professor J. C. Vassilicos for his insightful comments on subsection 4.1.



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
