# Peer review of "Reynolds-number dependence of turbulence enhancement on collision growth"

_Atmospheric Chemistry and Physics, 2016_

## Referee Comment (RC1) · Anonymous Referee #1 · 7 Mar 2016

**1   General Comments**

This article is concerned with the Reynolds-number dependence of the effect of turbulence on the collision statistics of cloud droplets. It updates a previously proposed empirical coagulation kernel proposed by Onishi et al. by using DNS data at higher Reynolds numbers, and compares the results to those of another model by Ayala and Wang. Theoretically deriving a coagulation kernel for such a problem is extremely difficult, and so one is forced to construct models for droplet collisions empirically, at least until the theoretical work matures sufficiently. Since the empirical models are not firmly grounded theoretically, one can never be really surely how they will extrapolate to larger Reynolds numbers, and so such studies such as in the present article are needed. The influence of turbulence on cloud droplet collisions and the effect that Reynolds number

has on this is certainly of interest to ACP.

The article is quite well written, and states the issues and objectives of the study clearly. However, one of the criticisms I have is that in several cases things are not explained in sufficient detail and that justification for the approximations invoked is not always given (discussed more in the next section of my review). The article does however address the questions that the study set out to consider, and provides useful results concerning how the effect of turbulence on droplet collisions might vary as one goes to higher and higher Reynolds numbers. The article also gives a helpful discussion and comparison of the two empirically based coagulation models, showing where they work well, and where they need to be improved.

There are, however, several important issues that the authors need to address before I can recommend the paper for publication in ACP, and these are explained below.

**2   Specific Comments**

- The authors do not sufficiently discuss, from a physical perspective, how/why changing the Reynolds number might change the collision behavior. The authors should include a discussion, based on modern results (e.g. see Extreme events in computational turbulence, Proc. Natl. Acad. Sci. USA, 2015, Yeung et al.), of how turbulence changes structurally/statistically when the Reynolds number increases, and how this might affect the collision behaviour.

- Two papers recently appeared on the arXiv by Ireland et al. (arXiv:1507.07026 and arXiv:1507.07022) that use DNS to consider, from a fundamental perspective, how changing the Reynolds number of the turbulence affects particle collisions in turbulence. The authors of the present article should comment on how their results and conclusions compare with those of Ireland et al. This is partic-

ularly important since Ireland et al. suggest that the effects of Reynolds number on the collisions may not be so important.

- In the DNS simulations, periodic boundary conditions are used and the particles are subject to gravity. In Ireland et al. (arXiv:1507.07022) it is shown that the simulation box needs to be quite large to avoid errors associated with the settling particles looping through the periodic box during the integral timescale of the turbulence. The results of Ireland et al. seem to show that for simulation domains of the size used in the DNS in the present article ($2\pi L_0$) such errors could be significant. Can the authors comment on this? How might such errors influence the results and conclusions of the present article?

- In section 2, I could not see any explanation regarding what particle equations of motion these collision kernels relate to?

- Regarding equation 26, the authors make no mention of the validity of such an equation of motion. What about nonlinear drag effects, or finite particle sizes for the larger $St$ particles?

- Regarding equation 10; presumably this model was derived for the case without gravity. Recently published results show that for $St \geq \mathcal{O}(1)$, the scaling of the RDF power law exponent with $St$ differs significantly with and without gravity (with gravity it varies vary slowly with increasing $St$ for $St \geq \mathcal{O}(1)$, and definitely not like $St^{-2}$). Could the authors comment on this?

- Where does equation 21 come from? What are the assumptions behind this?

- Can the authors include error bars on some of their plots? This would help to show the statistical significance of the argued Reynolds number dependencies of the collision statistics.

---

## Referee Comment (RC2) · Anonymous Referee #2 · 12 Apr 2016

**Review on the manuscript Atmos. Chem. Phys. Discuss. acp-2016-19, Reynolds-number dependence of turbulence enhancement on collision growth, by Ryo Onishi and Axel Seifert**

In this paper, the authors intend to compare two descriptions of turbulent collision kernels (Onishi vs Ayala-Wang) in evaluating the Reynolds number dependence of the turbulence enhancement (relative to the base gravitational kernel). They provided additional geometric collision statistics ($g_{11}$) for $R_\lambda$ up to 1140, which could be valuable. They also provide a model to explain the Reynolds number dependence of the radial distribution function using dissipation intermittency. However, their DNS data for turbulent collision efficiency and droplet-droplet hydrodynamic interactions, due to their BiSM approximation, may underestimate the collision efficiency particularly for the monodisperse case (also see comment 10 below). It should be noted that, even the more rigorous HDNS model developed in Ayala et al. (2007) does not treat short-range interactions correctly (e.g., see Rosa et al., J. Comp. Phys., 230, 8109-8133). In other words, while the authors' DNS data for the geometric collisions and the Reynolds number dependence of $g_{11}$ are carefully obtained, their DNS data for turbulent collision efficiency may not be accurate. This brings a question if their LCS results are accurate and can be used as a benchmark in Figs. 9 and 10.

The paper may be published, if the following issues can be considered and clarified in the revision.

1. Page 7, Eq. (16), what $\rho_{12}$ expression is used? Please provide the detail. $St_{\max}$ is not defined.

2. Eq. (20) is missing a description for $T_L$.

3. The rationale for the St-dependence in the two limits of small and large $St$ should be provided. At large $R_\lambda$, $St_a$ can even be large than one. I think the $St^2$ dependence, derived from small $St$, would not apply.

4. Eq. (21): In the limit of very large $V_{p,\infty}$, the fluid time scale seen by a sedimenting particle approach $L_f/V_{p,\infty}$, where $L_f$ is the longitudinal spatial velocity correlation length (e.g, Wang and Stock, J. Atmos. Sci. 50:1897-1913, 1993). Then $\theta_{i,sed}$ becomes $\tau_p V_{p,\infty}/L_f$. Eq. (21) does not seem to reduce to this result.

5. Page 8, the last sentence following Eq. (22) is confusing in two regards. First, clarify what the notation $\langle |w_r| \rangle$ is. If it is already averaged as the angle brackets usually mean, it should not have a distribution. Second, for the case of gravity only, the distribution of $|w_r|$ can be derived (see, e.g., Wang et al. J. Atmos. Sci. 63, 881 - 900.) and it is not Gaussian.

6. Page 9, first paragraph. The meaning of $E_{c,PKS01}$ needs to be clarified. Is this the collision efficiency for gravitational collision from PKS01? Other places in the paper, $E_c$ is used to indicate the collision efficiency for turbulent collision.

7. The dissipation ratio in Eq. (33) is more like $\langle(\partial u_1/\partial x_1)^4\rangle/(\langle(\partial u_1/\partial x_1)^2\rangle)^2$, so it is not flatness.

8. The symbols in Fig. 2 need to be better explained. Why are there six different types of symbols and what do they represent?

9. Fig. 4, the large value for the monodisperse case in Ayala model is due to large collision efficiency. The reference DNS data is based on the binary based superposition method (BiSM). Wang et al. (2008) found that the turbulent collision efficiency depends on the liquid water content, implying that the long-range multiple-droplet hydrodynamic interactions are important. I wonder if BiSM will encounter systematic error when simulating turbulent collision efficiency for the monodisperse case, so the reference DNS data and LCS data can no longer be used as the benchmark.

10. Figs. 5 to 8: When the droplet radius is above 100 $\mu m$, droplet deformation and coagulation efficiency must be considered. I think the discussions in this paper should be focused on $a < 100\mu m$, due to the large number of assumptions involved.

---

## Author Comment (AC1) · 9 May 2016

We appreciate your positive and insightful comments. Below we answer all the questions one by one.

**(1) The authors do not sufficiently discuss, from a physical perspective, how/why changing the Reynolds number might change the collision behavior. The authors should include a discussion, based on modern results (e.g. see Extreme events in computational turbulence, Proc. Natl. Acad. Sci. USA, 2015, Yeung et al.), of how turbulence changes structurally/statistically when the Reynolds number increases, and how this might affect the collision behaviour.**

The physical perspective was discussed in Onishi and Vassilicos (2014) as described in subsection 4.1: Onishi and Vassilicos (2014) clarified that the Reynolds-number dependence of g11 observed for $1/3 < St < 1$ is due to internal intermittency of the three-dimensional turbulence. Onishi and Vassilicos (2014) proposed a plausible mechanism that can explain the Reynolds-number dependence for $1/3 < St < 1$ by defining the local St (St*), via the local flow strain rate, based on K62. As the Reynolds number increases, an increasing part of space is dominated by small St*, which would decrease the clustering effect ($g_{11}$). As the area of St*>1 cannot efficiently increase $g_{11}$, the extreme local strain rates cannot tip the balance and overcome the reduction in $g_{11}$ caused by the reduced values of local strain rates in most of the space. The proposed mechanism does not need the modern finding for the 'very extreme' events observed in Yeung et al. (2015), it just needs the K62 model for intermittency. As the physical perspective is fully discussed in Onishi and Vassilicos (2014), this manuscript avoids repeating it.

**(2) Two papers recently appeared on the arXiv by Ireland et al. (arXiv:1507.07026 and arXiv:1507.07022) that use DNS to consider, from a fundamental perspective, how changing the Reynolds number of the turbulence affects particle collisions in turbulence. The authors of the present article should comment on how their results and conclusions compare with those of Ireland et al. This is particularly important since Ireland et al. suggest that the effects of Reynolds number on the collisions may not be so important.**

As described in subsection 4.2.2 in Ireland et al. (arXiv:1507.07026), there is a significant discrepancy in Ireland's conclusions and ours. Our DNS shows a decreasing trend of the clustering effect over the range 81< $Re_\lambda$ <527 at St=0.4 and 0.6. However, Ireland et al. did not find such trend and concluded the Reynolds-number dependence of the clustering effect at low St is negligibly small. However, if we carefully look at

Figure 20(a) in Ireland et al. (arXiv:1507.07026), we can see a consistent decrease of the clustering effect at St=0.3, 0.5 and 0.7 when $Re_\lambda$ increases from 224 to 597. As Ireland et al. (arXiv:1507.07026) shows the RDF in log scale, the decrease looks very small. But in linear scales the decreasing trend can be visible as in Rosa et al. (2013) as well as in our previous DNS studies. This manuscript can settle this dispute as Figure 2 clearly explains that the Reynolds-number dependence is significant when we discuss the large Reynolds numbers as observed in turbulent clouds (although it would not be visible, particularly in log scales, when we discuss the limited range of $Re_\lambda$ <600).

- Reference: Rosa et al., Kinematic and dynamic collision statistics of cloud droplets from high-resolution simulations., New J. Phys., 15, 045032 (2013)

**(3) In the DNS simulations, periodic boundary conditions are used and the particles are subject to gravity. In Ireland et al. (arXiv:1507.07022) it is shown that the simulation box needs to be quite large to avoid errors associated with the settling particles looping through the periodic box during the integral timescale of the turbulence. The results of Ireland et al. seem to show that for simulation domains of the size used in the DNS in the present article (2$\pi$L0) such errors could be significant. Can the authors comment on this? How might such errors influence the results and conclusions of the present article?**

As noted in Woittiez et al. (2009) and discussed in Appendix A in Ireland et al. (arXiv:1507.07022), the periodicity may lead to errors for the settling particles with large St. Ireland et al. (arXiv:1507.07022) defined the critical St, $St_{crit}$, as $St_{crit} = Fr \frac{L}{l} \frac{u'}{u_\eta}$, where Fr is the Froude number (=$a_\eta/g$, where $a_\eta$ is the Kolmogorov-scale acceleration), $L$ (=$2\pi L_0$ in this study) is the domain size, $l$ is the integral scale and $u_\eta$ is the Kolmogorov-scale velocity. For St larger than $St_{crit}$, the periodicity problem may arise. Figs. 4, 9 and 10 are for settling particles. For those figures, we

have calculated Stcrit to check the periodicity problem. (i)For Fig. 4, $St_{crit}$=3.7, which corresponds to $r_{crit}$=75um; $r_{crit}$ is the radius of particle with St=$St_{crit}$. The two plots from DNS, which correspond to $r_2$=80um and 120um, exceeds $r_{crit}$. However, since the two plots are more or less similar with the gravitational (Hall) kernel values, the turbulent contribution would be small compared to the gravitational settling contribution. Thus the error due to the periodicity would not significantly affect the results. (ii)For Figs. 9(a) and 10(a), $r_{crit}$ are 50, 65 and 70um for $\epsilon$=100, 400 and 1000 cm$^2$/s$^3$, respectively. For Figs. 9(b) and 10(b) $r_{crit}$ are 65, 75, 85 and 90um for $Re_\lambda$=66.1, 127, 206 and 333, respectively. The enhancement factor $E_{turb}$, shown in Figs. 9 and 10, was evaluated by $t_{10\%}$, which is defined as the time required for a cloud to convert 10% of its cloud mass into rain category drops. The threshold between cloud and rain categories was set at $r$=40um. That is, 10% of particles, in mass and volume, are larger than 40um in radius at $t = t_{10\%}$ by definition. For example, according to the DNS results, 3% of particles are larger than 50um and only 0.9% of particles are larger than 60um at $t = t_{10\%}$. The percentage of particles that are larger than 50um in radius may have some impact on $t_{10\%}$ and consequently Eturb. In this sense, the plot for $\epsilon$=100 cm$^2$/s$^3$ in Figs. 9(a) and 10(a), whose $r_{crit}$ is 50um, may contain some error associated with the periodicity problem. However, since $E_{turb}$ for the plot is nearly unity indicating small turbulence enhancement, the periodicity problem does not change the present findings. Overall, the periodicity problem does not seem significant for the present manuscript, but it is worth mentioning. The above discussion has been added as Subsection 4.6: Periodicity influence.

**(4) In section 2, I could not see any explanation regarding what particle equations of motion these collision kernels relate to?**

Section 2 describes the collision statistics in general, irrespective of the governing equations of particle motions.
**(5) Regarding equation 26, the authors make no mention of the validity of such an equation of motion. What about nonlinear drag effects, or finite particle sizes for the larger St particles?**

The nonlinear drag effect is included in Eq.(26); f shows the nonlinear drag coefficient. The finite-size effect is, however, ignored. Accordingly, we have added the following sentence in the last part of the corresponding section:
*"It should be noted that Eq. (26), which adopts the point-particle assumption, is inaccurate for large St particles whose radii are not small enough compared to the Kolmogorov scale."*

**(6) Regarding equation 10; presumably this model was derived for the case without gravity. Recently published results show that for St$\geq$O(1), the scaling of the RDF power law exponent with St differs significantly with and without gravity (with gravity it varies vary slowly with increasing St for St$\geq$O(1), and definitely not like St$^2$. Could the authors comment on this?**

Yes, Eq(10) is for the case without gravity. The gravity can alter the clustering effect leading to some errors in our g(R) model. The gravity influence can be significant for large droplets. For the cloud system containing such large droplets, collisions due to the settling velocity difference would be more important than those due to turbulence. This can probably mask the insufficiency of our g(R) model. It actually did for the present work as shown in good agreements between our kernel model and DNS results. For the case without gravity, the results from Ireland et al. (arXiv:1507.07026) supports Eq. (10). It should be noted that Eq. (10) is the model for the RDF at contact, i.e., $g_{11}(x = 2r_1)$, not for the RDF, which is the function of the radial distance $x$. It

should be also noted that the power law exponent is not the only measure of the RDF at contact, and the coefficient $C_0$ is also the key parameter. Reade & Collins formulation leads to

$$g_{11}(r) = C_0(\frac{\eta}{r})^{C_1} \propto C_0 St^{-C_1/2}.$$

For example, if we look at the data for $Re_\lambda$=224 in Figure 22 in Ireland et al. (arXiv:1507.07026) $C_0$ and $C_1$ are 6 and 0.45, respectively, for St=2, and 4 and 0.3 for St=3. Substitutions of these values into Eq.(A1) yield $g_{11}$=5.1 for St=2 and 3.4 for St=3, leading to $\{g_{11}(St = 2) - 1\}/\{g_{11}(St = 3) - 1\}$=1.7. This value is not far from the prediction 2.25 from Eq. (10).

**(7) Where does equation 21 come from? What are the assumptions behind this?**

Onishi et al. (2009) derived Eq. (21). We modified the sentence that includes Eq. (21) into *"Onishi et al. (2009) modeled the enlargement of the relative particle relaxation time by gravity as* ..." The detail derivation is described in Onishi et al. (2009) and thus the manuscript avoids repeating it.

**(8) Can the authors include error bars on some of their plots? This would help to show the statistical significance of the argued Reynolds number dependencies of the collision statistics.**

Accordingly, we have added the error bars in Figures 2 and 4, and the corresponding explanation in the captions.

---

## Author Comment (AC2) · 9 May 2016

Thank you for your insightful comments. Below we answer all the questions one by one.

**(1) Page 7, Eq. (16), what r12 expression is used? Please provide the detail. Stmax is not defined.**

Zhou et al. (2001) proposed an empirical formulation for the correlation between the two concentration fields, based on their DNS results, as

$$\rho_{12} = 2.6 \exp(-St_{max}) + 0.205 \exp(-0.0206 St_{max})\frac{1}{2}[1 + \tanh(St_{max} - 3)].$$

[Figure]

The description for eq. (16) has been modified accordingly. $St_{max}$ is defined as $St_{max} = \max(St_1, St_2)$, i.e., the larger St of two different sized droplets, at Eq.(7).

**(2) Eq. (20) is missing a description for $T_L$.**

We used the formulation of $T_L = 0.4T_e$, where $T_e$ ($=u'^2/\epsilon$) is the large-eddy turnover time (Kruis and Kusters, 1997; Zhou et al., 2001). This information has been added in the revised manuscript.

**(3) The rationale for the St-dependence in the two limits of small and large St should be provided. At large $R_\lambda$, $St_a$ can even be large than one. I think the St$^2$ dependence, derived from small St, would not apply.**

Yes, $St_a$ can be larger than 1 at some large $R_\lambda$. But it does not mean that St$^2$ –dependence holds for St~1. For St~ $St_a$, $z_a$ defined by Eq. (12), and used in Eq. (11), becomes 0.5, leading to a break of the St$^2$ –dependence in our empirical parameterization.

**(4) Eq. (21): In the limit of very large $V_{p,\infty}$, the fluid time scale seen by a sedimenting particle approach $L_f/V_{p,\infty}$, where $L_f$ is the longitudinal spatial velocity correlation length (e.g, Wang and Stock, J. Atmos. Sci. 50:1897- 1913, 1993). Then $\theta_{i,sed}$ becomes $\tau_p V_{p,\infty}/L_f$. Eq. (21) does not seem to reduce to this result.**

Eq. (21) is consistent with the correlation by Wang and Stock for the limit of very large $V_{p,\infty}$. For $V_{p,\infty} \gg 1$, i.e., for $s_v \gg 1$, $\theta_{i,sed}$ becomes

$s_v\theta_i = (V_{p,\infty}/u_{rms})(\tau_{p,i}/T_L) \sim \tau_{p,i}V_{p,\infty}/L_f$ with assuming $T_L \sim T_i$ (e.g., Gouesbet et al. Phys. Fluids, 27, 827-837: 1984), where $T_i$ is the fluid integral scale, and $L_f \sim u_{rms}T_L$. Of course, as Wang and Stock (1993) pointed out, the assumption of $T_L \sim T_i$ is problematic. However, the problem would not be serious for this study since the particle velocity fluctuations become negligible and the turbulence effect on collisions become insignificant consequently.

**(5) Page 8, the last sentence following Eq. (22) is confusing in two regards. First, clarify what the notation $< |w_r| >$ is. If it is already averaged as the angle brackets usually mean, it should not have a distribution. Second, for the case of gravity only, the distribution of $|w_r|$ can be derived (see, e.g., Wang et al. J. Atmos. Sci. 63, 881 - 900.) and it is not Gaussian.**

The angle brackets denote the averaging procedure. We have removed some of the angle brackets. They were erroneous as you point out.

(original) *This simple form is exact if no clustering occurs and $< |w_r| >_{turb,sed}$ and $< |w_r| >_{grav}$ and follow Gaussian distributions.*
(after modification) *This simple form is exact if no clustering occurs and $|w_r|_{turb,sed}$ and $|w_r|_{grav}$ and follow Gaussian distributions.*

**(6) Page 9, first paragraph. The meaning of $E_{c,PKS01}$ needs to be clarified. Is this the collision efficiency for gravitational collision from PKS01? Other places in the paper, Ec is used to indicate the collision efficiency for turbulent collision.**

Yes, $E_{c,PKS01}$ is the collision efficiency (Ec) for gravitational collision from PKS01. We have modified the corresponding sentence accordingly. Hall (1980) also provides another set of values for Ec for gravitational collision: Ec,Hall. The notation '$E_c$' is used

for the collision efficiency in general in this manuscript.

**(7) The dissipation ratio in Eq. (33) is more like $\left\langle (\partial u_1/\partial x_1)^4 \right\rangle / \left\langle (\partial u_1/\partial x_1)^2 \right\rangle^2$, so it is not flatness.**

Yes. That's why Eq. (33) is written in an approximation form.

**(8) The symbols in Fig. 2 need to be better explained. Why are there six different types of symbols and what do they represent?**

As we cannot analytically calculate the integration in Eq. (38), we have to numerically calculate it to obtain $g_{11}$ for a certain combination of St and $Re_\lambda$. The six types of symbols correspond to six values of $Re_\lambda$ ($Re_\lambda$=100, 200, 400, 1000, 4000 and 10000). We have modified the corresponding explanation.

**(9) Fig. 4, the large value for the monodisperse case in Ayala model is due to large collision efficiency. The reference DNS data is based on the binary based superposition method (BiSM). Wang et al. (2008) found that the turbulent collision efficiency depends on the liquid water content, implying that the long-range multiple-droplet hydrodynamic interactions are important. I wonder if BiSM will encounter systematic error when simulating turbulent collision efficiency for the monodisperse case, so the reference DNS data and LCS data can no longer be used as the benchmark.**

Onishi et al. (2013) reported that BiSM is as reliable as the iterative superposition method (Ayala et al. 2007) for the typical liquid water content of 1 g/m³, while Wang

et al. (2008) investigated the collision efficiency for liquid water content ranging from 1 to 55 g/m$^3$, which are larger than the typical value in clouds (0.5-1 g/m$^3$). Although further investigation is needed, BiSM can be as reliable as the iterative method for cloud research.

It is not yet clear the source of the large discrepancy between the Ayala-Wang data and ours for the monodisperse case in Fig.4. We leave the investigation of this discrepancy for future study.

**(10) Figs. 5 to 8: When the droplet radius is above 100um, droplet deformation and coagulation efficiency must be considered. I think the discussions in this paper should be focused on $a <$ 100um, due to the large number of assumptions involved.**

It is true that we cannot make robust discussion for $a >$100um, and we do not actually discuss the droplets much larger than 100um. We have added the following note for readers in the end of Subsection 2.1:

"*Droplet deformation and coalescence efficiency, which this study ignores, affect the collision growth of droplets with $r >$100um, although such effects only become significant for droplets with $r >$500um. It would, therefore, lead to some errors if extending the present results to such large droplets.*"

---

## Referee Report (RR1)

**Review on the revised manuscript Atmos. Chem. Phys. Discuss. acp-2016-19, Reynolds-number dependence of turbulence enhancement on collision growth, by Ryo Onishi and Axel Seifert**

The revision addressed most of my questions. I only have some minor comments.

1. Eq. (21), in the limit of $s_v \to \infty$, $\theta_{i,sed} \to s_v \theta_i / \sqrt{3}$. Would the factor $\sqrt{3}$ present a problem?

2. Bottom of page 8, the statement "This simple form is exact ...." is certainly not true. Eq. (22) is one way to combine two mechanisms, and they are many possibilities.

---

## Author Response (AR2)

Reply to Referee #1

We appreciate your additional comments. Below we answer all the questions one by one.

(1)Concerning Figure 2.
(1-1)First, I realized that the text in the paper does not actually refer to this figure at any point (that I could see). I wonder whether the line below equation 42 is supposed to be referring to figure 2 and not figure 3?

>>Yes. It has been modified in the revised manuscript.

(1-2)Second, the results in Fig 2 suggest that in going from Re_lambda=O(100) to Re_lambda=O(10000), the RDF at contact can vary significantly if St is large enough. Yet the data for Re_lambda>O(1000) is based upon a model that attempts to capture the effects of intermittency, and not DNS data. Being a model, the results are open to question. On the other hand, in Ireland's recent JFM paper (Journal of Fluid Mechanics, Volume 796, June 2016, pp617 - 658, which is the published version of the arXiv paper I referred to in my first review), in Fig23, their DNS results show that the effect of Re_lambda on the RDF saturates as Re_lambda is increased. This is in contrast to the assertions of the present paper that claims in Fig 2 (using a model) that increasing Re_lambda continues to decrease the RDF up to Re_lambda=O(10000). The authors need to discuss this difference in detail. Since the Re_lambda dependence of the collision statistics is in fact the main topic of the present paper, the authors need to seriously consider how to explain the discrepancy between their results and those of Ireland, e.g. whether the effect of Re_lambda saturates or not etc. The authors should consider how the different numerical schemes employed in the two studies may be responsible for the discrepany etc.

>>We've included the data of Figure 23 of Ireland's paper (JFM, 2016, vol. 796, pp.617-658) in Figure 2 of the revised manuscript and revised the discussion for the figure at the end of Subsection 4.1. The updated figure clearly shows that both of ours and Ireland's results show a decreasing trend for St=0.6 for $Re\lambda$>200. The two DNS data are consistent (this should be emphasized), but the interpretation is different between our paper and Ireland's. We need further data for high Reynolds numbers to conclude which interpretation is actually correct. One thing to be noted is that we have proposed the plausible mechanism that the flow intermittency can cause the Re dependency (Onishi and Vassilicos, 2014 J. Fluid Mech.) and Figure 2 actually shows good agreement between the prediction and the DNS results.
Even the saturation of g11 for St=1 at $Re\lambda$=500 is fully consistent with our DNS data and the Onishi model of g11. The Onishi model does predict a

decrease in g11(St=1) for Reλ larger than 1000, though. Making predictions for the behavior outside of the range of available DNS data is, in our opinion, valuable, because it allows for a falsification (or validation) of the model when DNS at even higher Reλ becomes available. Obviously such predictions for high Reλ are also necessary to apply the results to atmospheric flows like clouds.

(2)Regarding section 2, the authors stated in response to my first review that section 2 is general, and is not specific to the equation of motion for the particles. But this is not correct, for example, as I pointed out in my initial review, equation 10 is only valid for monodisperse inertial particles, subject to Stokes drag only without gravity. So how then can section 2 be general, and not specific to the equation of motion for the particles?

>>We meant Subsection 2.1 is general. Sorry for our misunderstanding of your comment. Yes, Eq. (10) is developed based on the DNS for non-settling monodisperse particles. We have modified the sentence preceding to Eq. (10) as "*Onishi et al. (2015) proposed an original model for the clustering effect in monodisperse systems of non-sedimenting particles with Stokes' linear drag.*"

Reply to Referee #2

Thank you again for your insightful comments.

**(1) Eq. (21), in the limit of $s_v \to 1$, $\theta_{i,sed} \to s_v \theta_i / \sqrt{3}$. Would the factor $\sqrt{3}$ present a problem?**
>> The factor $\sqrt{3}$ comes from the underlying assumption that the particle velocity fluctuation is isotropic (i.e., the fluctuations in the vertical and horizontal directions are the same). This assumption is invalid for large $s_v > 4$ (Onishi et al., 2009 *Phys. Fluids*). But, it does not cause a problem for this study since the particle velocity fluctuations become negligible anyway and the turbulence effect on collisions become insignificant consequently.

**(2) Bottom of page 8, the statement "This simple form is exact ...." is certainly not true. Eq. (22) is one way to combine two mechanisms, and they are many possibilities.**
>> We have simply removed the sentence to avoid ambiguity.